# Incentive-driven transition to high ride-sharing adoption

David-Maximilian Storch [1], Marc Timme [1,2] & Malte Schröder [1✉]

Ride-sharing—the combination of multiple trips into one—may substantially contribute towards sustainable urban mobility. It is most efficient at high demand locations with many similar trip requests. However, here we reveal that people's willingness to share rides does not follow this trend. Modeling the fundamental incentives underlying individual ride-sharing decisions, we find two opposing adoption regimes, one with constant and another one with decreasing adoption as demand increases. In the high demand limit, the transition between these regimes becomes discontinuous, switching abruptly from low to high ride-sharing adoption. Analyzing over 360 million ride requests in New York City and Chicago illustrates that both regimes coexist across the cities, consistent with our model predictions. These results suggest that even a moderate increase in the financial incentives may have a disproportionately large effect on the ride-sharing adoption of individual user groups.

---

[1] Chair for Network Dynamics, Institute for Theoretical Physics and Center for Advancing Electronics Dresden (cfaed), Technical University of Dresden, Dresden, Germany. [2] Lakeside Labs, Lakeside B04b, Klagenfurt, Austria. ✉email: malte.schroeder@tu-dresden.de

Sustainable mobility[1–6] is essential for ensuring socially, economically, and environmentally viable urban life[7,8]. Ride-sharing (sometimes also referred to as ride-pooling) constitutes a promising alternative to individual motorized transport by private cars or single-occupant taxi cabs, currently dominating urban mobility[9]. In ride-sharing, one vehicle transports multiple passengers simultaneously by combining two or more trip requests with similar origin and destination. In contrast to analog on-street hailing of taxi rides, digital app-based ride-hailing services are especially suited to implement ride-sharing due to easy access to the information required to match different trips.

By combining different individual trips into a shared ride, ride-sharing increases the average utilization per vehicle, reduces the total number of vehicles required to serve the same demand[10] and thereby mitigates congestion and negative environmental impacts of urban mobility[11,12]. Hence, encouraging ride-sharing for trips that would otherwise be conducted in a single-occupancy motorized vehicle is preferable from a systemic perspective[12–15].

Previous research focused on developing algorithms to implement large-scale ride-sharing[16] as well as the potential efficiency gains derived from aggregating rides[9,17,18]. Recent analyses suggest that large-scale ride-sharing is most efficient in densely populated urban areas[9,10,17–19] since matching individual rides into shared ones without large detours becomes easier with more users, increasing both the economic and environmental efficiency as well as the service quality of the ride-sharing service[17,18,20]. Yet, if and under which conditions people are actually willing to adopt ride-sharing remains elusive[21–28]. In particular, it is unclear how to encourage an ever growing number of ride-hailing users to choose shared rides over their current individual mobility options[29–31].

In this article, we disentangle the complex incentive structure that governs ride-hailing users' decisions to share their rides—or not. In a game theoretic model of a one-to-many demand constellation we illustrate how the interactions between individual ride-hailing users give rise to two qualitatively different regimes of ride-sharing adoption: one low-sharing regime where the adoption decreases with increasing demand and one high-sharing regime where the population shares their rides independent of demand. Analyzing ride-sharing decisions from approximately 250 million ride-requests in New York City and 110 million in Chicago suggests that both adoption regimes coexist in these cities, consistent with our theoretical predictions. Our findings indicate that a small increase in financial incentives may disproportionately increase the adoption of ride-sharing for individual user groups from a low to a high-sharing regime.

## Results

**Contrasting ride-sharing adoption.** Currently, only a small fraction of people adopts ride-sharing even in high-demand situations, despite all its positive aspects[32]. For example, among more than 250 million ride-hailing requests served in New York City in 2019 less than 18% were requests for shared transportation[33]. Moreover, the city's ride-sharing activity varies strongly across different parts of the city, in particular at locations with a high number of ride-hailing requests (see Fig. 1): For instance, in the East Village and Crown Heights North the fraction of shared ride requests is relatively high, while it is low at both John F. Kennedy and LaGuardia airports, locations that would intuitively be especially efficient for sharing rides. Several other location throughout New York City as well as Chicago exhibit similarly contrasting ride-sharing adoption (see Supplementary Notes 1 and 2 for details). These findings hint at a complex interplay of urban environment, demand structure and

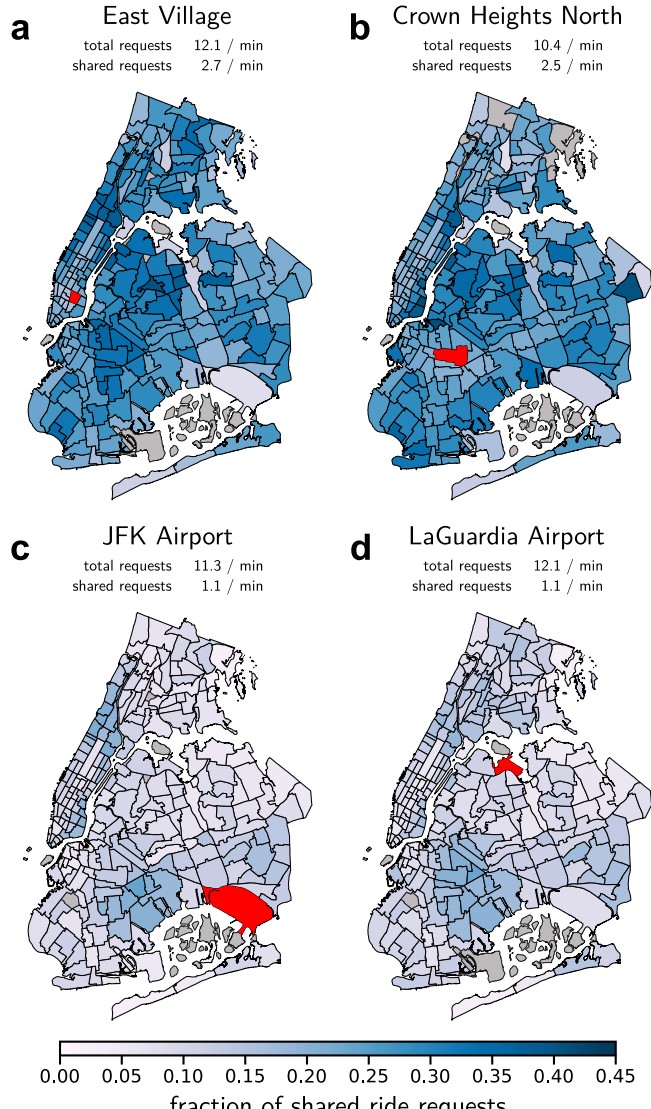

**a** East Village
total requests 12.1 / min
shared requests 2.7 / min

**b** Crown Heights North
total requests 10.4 / min
shared requests 2.5 / min

**c** JFK Airport
total requests 11.3 / min
shared requests 1.1 / min

**d** LaGuardia Airport
total requests 12.1 / min
shared requests 1.1 / min

0.00 0.05 0.10 0.15 0.20 0.25 0.30 0.35 0.40 0.45
fraction of shared ride requests

**Fig. 1 Contrasting ride-sharing adoption despite high request rate in New York City.** Fraction of shared ride requests from different origins (red) served by the four major for-hire vehicle transportation service providers in New York City by destination zone (January - December 2019)[33]. Gray areas were excluded from the analysis due to insufficient data (see Methods). The fraction of shared ride requests differs significantly by origin and destination with a complex spatial pattern across destinations, even though the average overall request rate is similar for all four origin locations. **a**, **b** Some areas, such as East Village and Crown Heights North, show a high adoption of ride-sharing services. **c**, **d** Despite a similarly high request rate, other locations, such as JFK and LaGuardia airports, show a significantly lower adoption of ride-sharing services.

socio-economic factors that govern the adoption of ride-sharing. To disentangle these complex interactions, we introduce and analyze a game theoretic model capturing essential features of ride-sharing incentives, disincentives as well as topological demand structure.

**Ride-sharing incentives.** The decision of ride-hailing users to request a single or a shared ride reflects the balance of three fundamental incentives[22,26]: financial discounts, expected detours as well as uncertainty about the duration of the trip, and the inconvenience of sharing a vehicle with strangers. Strong correlations between the adoption of ride-sharing and (in)direct

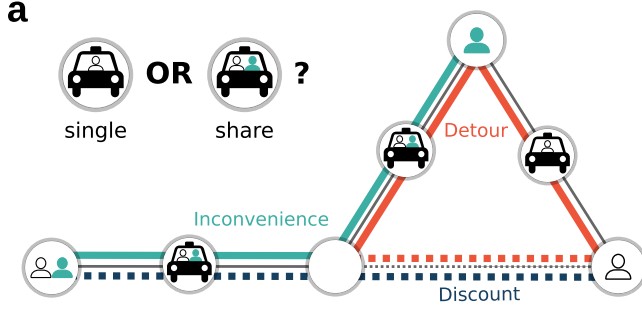

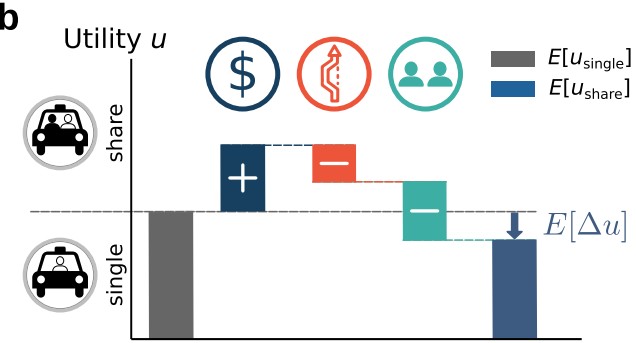

**Fig. 2 Trade-offs between incentives determine the decision to share a ride, or not. a** Shared rides offer advantages and disadvantages compared to single rides. On the one hand, they offer financial discounts typically proportional to the distance of a direct single ride (blue, dotted). On the other hand, rides shared with strangers may include detours compared to a direct trip to pickup or deliver these other passengers (orange, solid compared to dotted) and may be inconvenient due to other passengers in the car (e.g., loss of privacy or less space, green). **b** The decision to book a shared ride depends on the balance of all three factors. If the expected utility difference $E[\Delta u] = E[u_{\text{share}}] - E[u_{\text{single}}]$ between a shared and a single ride is positive, the financial discounts overcompensate detour and inconvenience effects; users share. If $E[\Delta u]$ is negative (as illustrated), users prefer to book single rides.

measures of the three incentives (see Supplementary Notes 1 and 2, including Supplementary Figs. 5 and 7) confirm the importance of these incentives found in detailed empirical studies of ride-sharing user experiences as well as focus group interviews[21,23–25,27,28,34–38]. Together, discounts, detours and inconvenience affect the ride-sharing adoption as follows (Fig. 2):

Discounts: Ride-sharing is incentivized by financial discounts granted on the single ride trip fare, partially passing on savings of the service cost to the user. Often, these discounts are offered as percentage discounts on the total fare such that the financial incentives $u_{\text{fin}}^{\text{share}} > 0$ are approximately proportional to the distance or duration $d_{\text{single}}$ of the requested ride, $u_{\text{fin}}^{\text{share}} = \epsilon\, d_{\text{single}}$, where $\epsilon$ denotes the per-distance financial incentives. In many cases, these discounts are also granted if the user cannot actually be matched with another user into a shared ride[39,40].

Detours: Potential detours $d_{\text{det}}$ to pickup or to deliver other users on the same shared ride discourage sharing. The magnitude of this disincentive $u_{\text{det}}^{\text{share}} < 0$ increases with the detour $d_{\text{det}}$.

Inconvenience: Sharing a ride with another user may be inconvenient due to spending time in a crowded vehicle or due to loss of privacy[22,24,25]. This disincentive $u_{\text{inc}}^{\text{share}} < 0$ scales with the distance or duration $d_{\text{inc}}$ users ride together.

In the following we take $u_{\text{det}}^{\text{share}} \propto d_{\text{det}}$ and $u_{\text{inc}}^{\text{share}} \propto d_{\text{inc}}$, describing the first order approximation of these disincentives

and matching the linear scaling of the financial incentives with the relevant distance or time.

These incentives for a shared ride describe the difference $\Delta u$ in utility compared to a single ride or another mode of transport. The overall utility of a shared ride is then given by

$$\begin{aligned} u_{\text{share}} &= u_{\text{single}} + \Delta u \\ &= u_{\text{single}} + u_{\text{fin}}^{\text{share}} + u_{\text{det}}^{\text{share}} + u_{\text{inc}}^{\text{share}} \quad (1) \\ &= u_{\text{single}} + \epsilon\, d_{\text{single}} - \xi\, d_{\text{det}} - \zeta\, d_{\text{inc}} \end{aligned}$$

where the utility $u_{\text{single}}$ for a single ride describes the benefit of being transported, as well as the cost and time spent on the ride. The factors $\epsilon$, $\xi$ and $\zeta$ denote the user's preferences. By rescaling the utilities (measuring in monetary units), $\epsilon$ directly denotes the relative price difference between single and shared rides whereas $\zeta$ and $\xi$ quantify the importance of inconvenience and detours relative to the financial incentives (see Supplementary Note 3 for details).

For a given origin-destination pair with fixed single ride distance $d_{\text{single}}$, financial incentives are constant for a given discount factor $\epsilon$. In contrast, detour and inconvenience contributions depend on the destinations and sharing decisions of other users. Their magnitude depends on where these users are going and on the route the vehicle is taking for a shared ride (see Methods). The decision to share a ride is determined by the expected utility difference (see Fig. 2)

$$E[\Delta u] = E[u_{\text{share}}] - E[u_{\text{single}}] \quad (2)$$

where $E[\,\cdot\,]$ signifies the expectation value over realizations of other users' destinations and sharing decisions conditional on one's own sharing decision.

**Ride-sharing coordination game on networks.** To understand how these incentives determine the adoption of ride-sharing, we study sharing decisions in a stylized city network[41] with a common origin $o$ in the center (e.g., a central downtown location) and multiple destinations $d$ (illustrated in Fig. 3). Two rings define urban peripheries equidistant from the city center. Branches represent cardinal directions of destinations. Requests for shared rides will only be matched along adjacent branches, if the shared ride reduces the total distance driven to deliver the users and to return to the origin compared to single rides, consistent with a profit-maximizing service provider. Pairing at most two users who request a shared ride, the problem of matching shared ride requests reduces to a minimum-weight-matching with an efficient solution, eliminating the influence of heuristic matching algorithms[16,18] (see Methods for details).

In this one-to-many setting, users requesting a shared ride would only share a ride if they make their requests within some small time window $\tau$. Therefore, we consider a game with $S = s\,\tau$ users traveling to a uniformly chosen destination location, where $s$ denotes the average request rate. These users have the option to book a single ride or a shared ride at discounted trip fare. Their decision to share depends on their expected utility difference $E[\Delta u(d)]$ [Eq. (2)], now depending on their respective destination $d$. Users observe their respective utility differences $E[\Delta u(d)]$ over a number of rides and adapt their sharing decision to maximize their expected utility. Eventually, users' sharing decisions converge to the equilibrium probabilities $\pi^*(d)$, reflecting an optimal response that maximizes the utility of users going to destination $d$ (see Methods for details).

At fixed discount $\epsilon$ and preferences $\zeta$ and $\xi$ ride-hailing users may decrease their overall adoption of ride-sharing $\langle\pi^*\rangle$ as the total number $S$ of users increases (see Fig. 3a, blue), even though ride-sharing becomes more efficient with higher user numbers.

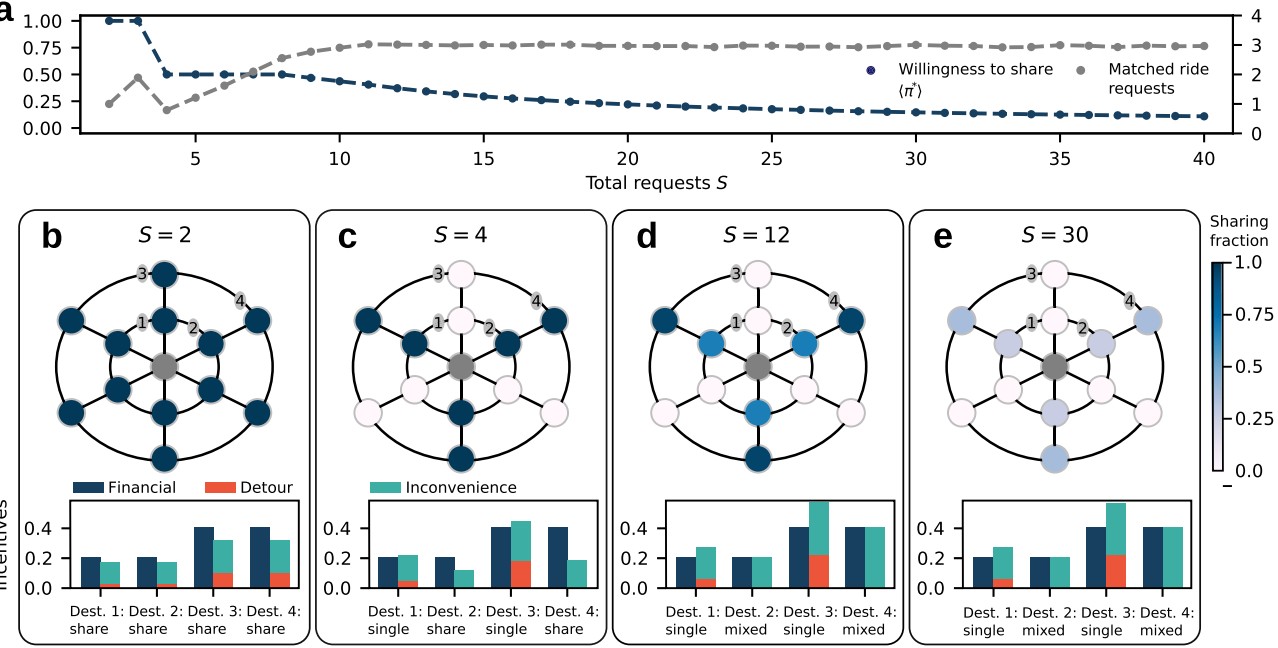

**Fig. 3 Adoption of ride-sharing decreases with request rate.** In a stylized city topology (panels **b**–**e**) users request transportation from a single origin (gray) to destinations in the city periphery homogeneously (results are robust for alternative settings, see Supplementary Note 4). **a** The global equilibrium adoption of ride-sharing decreases as the number of users increases (blue) while the number of actually shared rides becomes constant (gray). The kink for $S = 3$ is an artefact related to the small and odd number of requests and matching of exactly two requests per vehicle such that one request can never be paired (see Supplementary Note 3 for details). **b**–**e** As the number of users increases, ride-sharing adoption decreases and a sharing/non-sharing pattern emerges around the origin (top), resulting from the equilibrium incentive balance (bottom, illustrated for the numbered destinations) and possible matching constellations. Requests for shared rides are only matched when traveling to the same or to neighboring branches when the combined trip and return is shorter than the sum of individual trips. With few requests ($S = 2$, panel **b**), all users request a shared ride. The expected detour and inconvenience is small since it is unlikely to be matched with another user. As the number of users increases ($S = 4$, panel **c**), half of the destinations stop sharing in an alternating sharing/non-sharing pattern around the origin. In this configuration, users requesting a shared ride never suffer any detour while users that do not share are disincentivized from doing so due to their high expected detour (compare bottom part of panel **c**). For high numbers of users ($S = 12$ and 30, panels **d** and **e**), the probability to be matched with another user when requesting a shared ride increases and the financial incentives cannot fully compensate the expected inconvenience. The adoption of ride-sharing decreases until the financial incentives exactly balance the expected inconvenience (panels **d** and **e**, bottom). Illustrated here for financial discount $\epsilon = 0.2$ and inconvenience and detour preferences $\zeta = 0.3$ and $\xi = 0.3$.

Here $\langle \cdot \rangle$ denotes the average over all destinations $d$. While for small request rates everybody is requesting shared rides (Fig. 3b), a distinctive sharing/non-sharing pattern emerges along the branches of the city network upon higher demand (Fig. 3c, d), before the adoption of ride-sharing eventually fades out for high request rates, $S \gg 1$ (Fig. 3e). This observation offers a novel perspective on the prevalent conclusion that increased demand improves the shareability of rides[9,18]. While more rides are potentially shareable, less people may be willing to share them.

The underlying incentives explain this phenomenon: Ideally, a user wants to book a shared ride (financial incentive) but without actually sharing the ride (inconvenience and detour). This discrepancy is consistently observable also in public vocalization of user sentiment about shared ride experiences[21,24,28], and exemplarily summarized by the user quote 'Every time I take a [shared ride] and it ends up being just me the entire ride I feel like a genius'[27]. The expected detour and inconvenience mediate a repulsive interaction between the sharing decisions of ride-hailing users, turning ride-sharing decisions into a complex anti-coordination game. For small request rates, i.e., small numbers of concurrent users $S$, the probability $p_{match}(d)$ for a user with destination $d$ to be matched with other users is low (see Fig. 3a, gray). Consequently, the expected detour $E[d_{det}(d)] = p_{match}(d) E[d_{det}(d) \mid match]$ is also small (analogously for the inconvenience). As illustrated in Fig. 3b, bottom, financial incentives outweigh the expected disadvantages of ride-sharing

such that everybody is requesting shared rides, $\pi^*(d) = 1$ for all destinations $d$, but is only rarely matched with another user. As the number of users $S$ increases, the provider can pair ride requests more efficiently given constant sharing decisions, $\partial p_{match}(d)/\partial S > 0$, resulting in more requests that are actually matched with another user (see Fig. 3a). Consequently, the expected detour and inconvenience also increase. However, instead of reducing the average adoption of ride-sharing homogeneously across all destinations, neighboring destinations adopt opposing sharing strategies (see Fig. 3b). In this sharing pattern, only destinations in identical cardinal direction can and will be matched into a shared ride, minimizing the detours for shared requests and simultaneously disincentivizing other users to start sharing due to high expected detours (Fig. 3c–e bottom). As the number of users $S$ increases further, the probability $p_{match}(d)$ would also increase at given sharing adoption $\pi(d)$. This leads to an adoption of mixed sharing strategies where the financial discounts and the expected inconvenience are exactly in balance (Fig. 3d, e). Further numerical simulations demonstrate that this transition robustly exists also for heterogeneous demand distribution across the destinations, asymmetric street network topologies modeled by different origin locations within the network, for stochastic utility functions and imperfect information, as well as under different matching strategies by the service provider (see Supplementary Note 4 with Supplementary Figs. 10–14, 17).

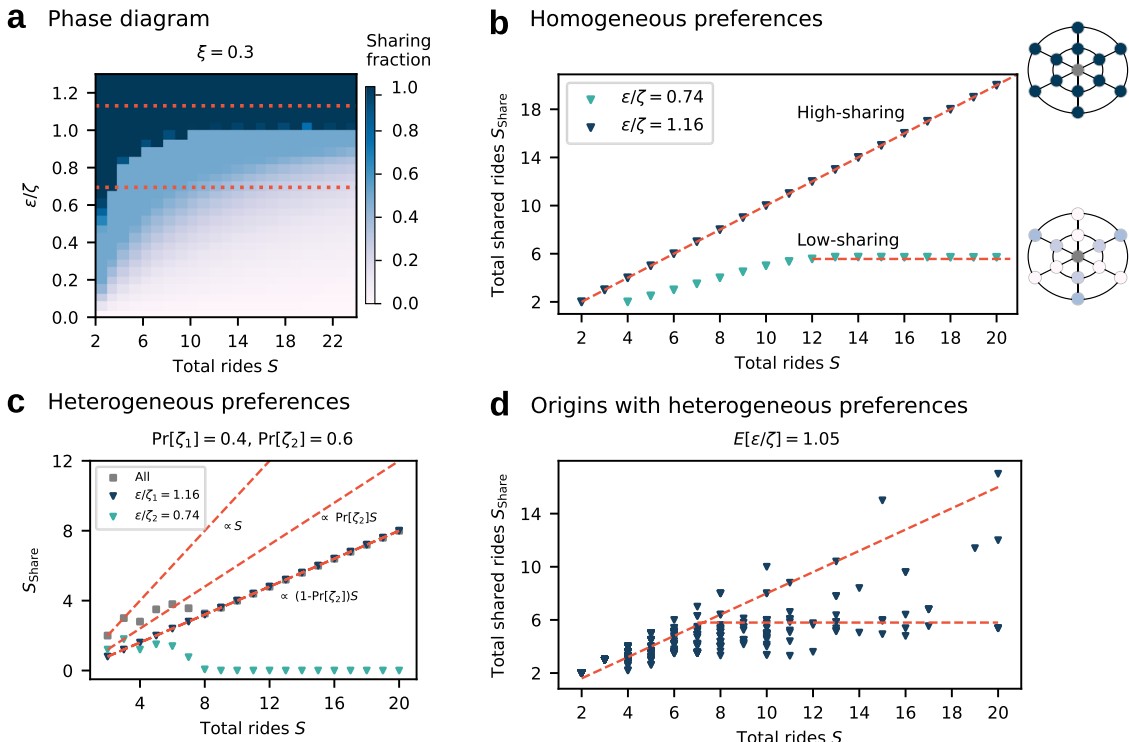

**Fig. 4 Transition from low to high ride-sharing adoption. a** Phase diagram of the fraction of shared rides $S_{share}/S$ for different relative importance of financial and inconvenience incentives $\epsilon/\zeta$. Ride-sharing is adopted dominantly if the financial discount fully compensates the expected inconvenience, $\epsilon/\zeta$ > 1 (high-sharing, dark blue). Otherwise, the total number of shared ride requests saturates and the overall adoption of ride-sharing decreases with increasing number of users $S$ (low-sharing, compare Fig. 3a). In the limit of infinitely many requests $S \to \infty$ the transition becomes discontinuous (see Supplementary Note 3). **b** Qualitatively different sharing behavior emerges for different relative incentives $\epsilon/\zeta$ (compare red lines in panel **a**). When $\epsilon/\zeta$ > 1 all users request shared rides ($S_{share} = S$, dark blue triangles). When $\epsilon/\zeta$ < 1, the system is in a low-sharing regime where users request shared rides at low numbers of users $S$ but the number of shared ride requests saturates and becomes constant at high $S$ ($S_{share}$, light green triangles). **c** Hybrid states of high- and low-sharing adoption may emerge if users with heterogeneous preferences $\epsilon/\zeta$ mix and interact. A fraction of users (for whom $\epsilon/\zeta$ > 1) is in the high sharing regime (blue). The others (green, for whom $\epsilon/\zeta$ < 1) decrease their ride-sharing adoption as the overall demand increases, consistent with the prediction for homogeneous user preferences (panel **b**). Macroscopically, the system exhibits partial ride-sharing adoption (gray). **d** The superposition of different realizations of this partial ride-sharing adoption represents the expected outcome in a city with multiple origins, each with heterogeneous preference distributions and demand (see Methods for parameters and Supplementary Note 4 for simulation details). While the macroscopic state suggests partial ride-sharing adoption, individual origins and user groups split into a mix of low- and high- sharing states, following the fundamental adoption regimes from the basic model.

Naturally, if the discount $\epsilon$ is sufficiently large such that the financial incentives completely compensate the expected inconvenience, $\epsilon > \zeta$, all users share also in the high request rate limit, $S \to \infty$. In this limit, $d_{single} = d_{inc}$ as detours disappear, $E[d_{det}] \to 0$, due to an abundance of similar requests. This transition is robust to changes of the model details, though under different matching strategies where detours remain possible in the high demand limit (see Supplementary Fig. 17), the financial incentives required to achieve high sharing adoption may be larger.

Figure 4 a summarizes these results in a phase diagram for the ride-sharing decisions as a function of financial discounts per inconvenience tolerance, $\epsilon/\zeta$, and number of users $S$, illustrating under which conditions the users adopt ride-sharing (high-sharing regime) and under which conditions the users only share partially or not at all (low-sharing regime).

For fixed values of financial discounts $\epsilon$ relative to the inconvenience preference $\zeta$ of the users, different behavior emerges (Fig. 4a): If $\epsilon/\zeta$ is sufficiently large ($\epsilon/\zeta$ > 1), the system is in the high-sharing state and all users request a shared ride ($S_{share} = S$). Otherwise ($\epsilon/\zeta$ < 1), the system transitions from the high- to a partial and finally to the low-sharing state (compare Fig. 3). Figure 4b illustrates the scaling of $S_{share}$ in both

cases as $S$ increases. In the low-sharing regime, $S_{share}$ eventually becomes constant for large $S$, such that $S_{share}/S \to 0$ as $S \to \infty$ (compare Fig. 3a). This implies a discontinuous phase transition between low-sharing and high-sharing regimes for large $S$ when the financial incentives exactly balance the inconvenience, $\epsilon_c/\zeta_c = 1$ (see Supplementary Note 3 and Supplementary Fig. 9).

For heterogeneous preferences within the population (e.g., different preferences of the individual users requesting rides from the same location) the transition robustly persists per user type. If $\epsilon/\zeta_i$ < 1 for parts of the local ride-hailing users, identified by their destination and preferences, these individuals transition from high- to low-sharing as the demand $S$ increases. The other part of the population, for whom $\epsilon/\zeta_i$ > 1, remains in the high-sharing state. Macroscopically, the system appears to be in a partial-sharing state even at very high demand (compare Fig. 4c), but in fact subsets of the population adopt opposing sharing strategies. The state of ride-sharing adoption across a city, i.e. across different origins each with a different distribution of inconvenience parameters and demand for rides $S$, is described by a superposition of these mixed states (see Fig. 4d). Macroscopically, the system may appear to be in a hybrid state of partial- and low-sharing adoption, even when the aggregate population on average

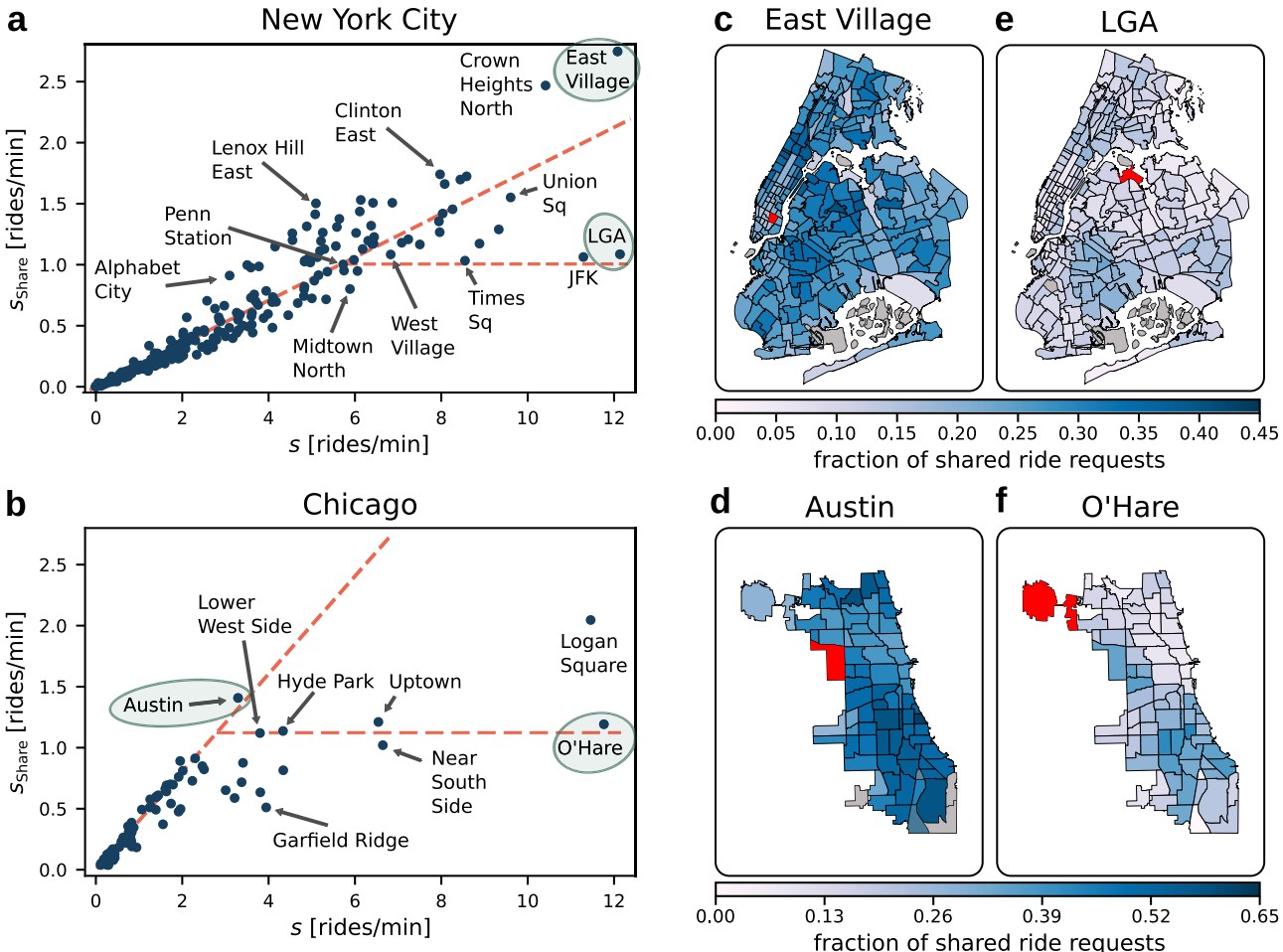

**Fig. 5 Ride-sharing adoption in New York City and Chicago is consistent with the predicted high- and low-sharing regimes. a, b** Sharing decisions for New York City and Chicago (blue dots) distribute between the two branches corresponding to the high- and low-sharing regime, consistent with the model predictions under heterogeneous user preferences (compare Fig. 4). At low request rates, the number of requests for shared rides increases linearly with the total number of requests (compare red diagonal). At high request rates, the sharing decisions differ between locations (compare Figs. 1 and 4, see also Supplementary Note 1 and 2). As inconvenience preferences $\zeta$ are naturally heterogeneous in the cities, adoption is in a hybrid low/high-sharing state. **c–f** Ride-sharing adoption is consistently higher across destination zones in the high-sharing regime compared to the low-sharing regime. The predominantly linear increase of the number of shared rides in New York City as demand increases suggests broadly sufficient financial compensation of sharing disutilities, or, alternatively a very broad range of user preferences, leading to a stable fraction of ride-sharing adoption. However, the slope of the high-sharing branch indicates that only about 20% of ride-hailing users consider ride-sharing as an option. While about 40% of requests are shared in the high-sharing regime in Chicago, this potential is largely not realized. The available data points at locations with relatively high request rate indicate a growth with the request rate that is much weaker than on average for the entire data set, or even absent, consistent with the low-sharing regime observed in our model. Seven large downtown zones in Chicago with up to 50 requests per minute (not shown) fall in between the high- and low-sharing state, likely representing the average of sharing behavior over a diverse population of ride-hailing users as expected for users with heterogeneous preferences (see Supplementary Fig. 6 for details).

satisfies $E[\epsilon/\zeta] > 1$ (see Methods, Supplementary Note 3 and Supplementary Figs. 15–16 for simulation details).

**Ride-sharing activity in New York City and Chicago**. Ride-sharing adoption across different parts in New York City (taxi zones) and Chicago (community areas), illustrated in Fig. 5 (see Methods and Supplementary Notes 1 and 2 for details), matches the qualitative sharing behavior expected for multiple origins with heterogeneous preferences and demand (compare Fig. 4d and Supplementary Note 4).

At locations with a low request rate $s$, the number of shared ride requests increases approximately linearly with more requests, $s_{\text{Share}} \sim s$. Though even in the low demand limit, the ride-sharing adoption in New York City and Chicago, corresponding to the diagonal branches in Fig. 5a, b, is below 100% (approximately

20% in New York City and 40% in Chicago). In terms of our ride-sharing game, the remaining fraction of requests for single rides may correspond to a user group with high relative importance of inconvenience compared to financial incentives, $\epsilon/\zeta \ll 1$, or that otherwise does not consider sharing as an option. In this interpretation, the smaller value for New York City is consistent with a large fraction of high-income and business customers in Manhattan who likely place a higher value on convenience than financial incentives.

At higher request rates, sharing decisions differ by origin zone and split between low and partial sharing states (compare Fig. 1). In New York City (Fig. 5a), Crown Heights North and East Village exhibit a relatively high ride-sharing adoption in line with that observed in low demand zones, indicating $\epsilon$ is sufficiently large to compensate the expected inconvenience and detour effects for a significant fraction of the users. Other origins with a

similarly high request rate, such as JFK and LaGuardia airports, do not follow this trend and exhibit a smaller number of shared ride requests. In terms of our model, we expect that $s_{\text{Share}}$ has largely saturated in these zones and the given financial incentives do not outweigh the perceived inconvenience of ride-sharing. In particular at the airports, it seems plausible that financial incentives for ride-sharing are less important to users in the context of already costly plane tickets. In Chicago (Fig. 5b), we find high-demand zones with an approximately constant number of shared ride requests, consistent with the low-sharing regime (horizontal branch $s_{\text{Share}} = $ const. in our model). In contrast, no zones with high demand show the same, relatively high ride-sharing adoption as zones with low demand. Some large downtown zones in Chicago with up to 50 requests per minute fall in the partial sharing regime expected for zones that effectively aggregate sharing decisions over a broad distribution of user preferences.

## Discussion

Ride-sharing bears a large potential in the transition towards more sustainable mobility[9,17]. Yet, it remains poorly understood how to unlock this potential due to the complex interplay of demand patterns, matching algorithms, available transportation options, urban environments and the relevant incentive structure governing the adoption of shared rides. We have introduced a game theoretical model capturing incentives for and against ride-sharing from a user perspective, reflecting the major incentives found in empirical studies of users' ride-sharing experience[21,23–28]. The model offers mechanistic insight into the collective effect of these incentives on individual ride-sharing decisions, unveils a discontinuous transition towards high overall ride-sharing adoption, and consistently explains the qualitative adoption of ride-sharing observed from 360 million empirical trip records from New York City and Chicago.

The ratio of financial discounts to inconvenience preferences acts as the control parameter in the model, separating two disparate regimes of ride-sharing adoption: one where the number of shared rides increases as the overall demand for rides increases (high-sharing regime) and one where it saturates (low-sharing regime), despite more efficient matching options and less detour as demand increases. Both regimes are separated by a regime with partial ride-sharing adoption that disappears in the high-demand limit. These results complement the finding of increased potential shareability of rides in high-demand settings[9,17] and may help to increase the service adoption to realize the full potential of ride-sharing under these conditions.

For homogeneous preference types across the user base, the adoption switches abruptly from the low adoption to the high adoption regime with a small change of the financial incentives and the transition between the two regimes becomes discontinuous (see Supplementary Note 3 for a mathematical proof). For heterogeneous preference types, as naturally expected in real cities with diverse population groups, the discontinuous transition robustly persists per user type. Macroscopically, however, heterogeneous preferences may induce a broad variance in the sharing adoption and yield mixed sharing decisions between the high- and -low sharing limit, blurring the abrupt transition towards high ride-sharing adoption as financial incentives increase. In line with our model predictions under spatially heterogeneous preferences, ride-sharing adoption observed in 360 million ride-sharing decisions from New York City and Chicago is broadly distributed across the cities, bounded between the high- and low-sharing regime (compare Figs. 4 and 5). Hence, the results above provide a consistent theoretical model and offer a possible explanation of qualitative features of ride-sharing

adoption in urban environments, based on empirical model ingredients. The mechanisms captured by the model are independent of details of the incentive structure, utility functions, or matching and service scheme applied by the provider. We illustrate this robustness for a wide range of different conditions beyond those illustrated in Fig. 3, including non-symmetric city topologies, heterogeneous demand distribution across possible destinations, noisy or imperfect information or decision-making, different strategies for matching rides, as well as for different simulation parameters (see Supplementary Note 4). However, deriving specific quantitative predictions from the model would require more detailed knowledge about users' preferences beyond the linear utility function assumed in our model. While the linear scaling in our model captures the basic features of the interactions, other models commonly assume a threshold dynamic to describe the impact of detour[9,17]. In addition, correlations in the demand structure and non-local matching of rides with different origins as well as the interplay between different service providers may also affect ride-sharing adoption. Similarly, the heterogeneity of ride-sharing adoption across different parts of the cities, expected in the low-sharing regime, seems to be dominated by socio-economic factors rather than by the pattern formation dynamics observed in our model network[42].

Future research may investigate in more detail the impact of inconvenience on the adoption of ride-sharing, but also extend the analysis to additional factors such as users' sustainability attitudes, explicit risk aversion in the light of detour uncertainty, or mode choices with regard to public transportation alternatives. Our model description may already provide a theoretical framework for many of these factors influencing ride-sharing adoption on an aggregate level. For example, sustainability or uncertainty preferences to first approximation scale with the additional distance driven and may thus be effectively described by the detour preference. Similarly, alternative public transport options may be captured by modifying the effective financial discount and relative inconvenience preferences for individual destinations.

The sharp transition to high-sharing adoption predicted by our model for any given set of preferences of a user, suggests that even a moderate increase of financial incentives or a small improvement in service quality may disproportionately increase ride-sharing adoption of user groups currently in the low-sharing regime under a broad range of conditions. On the other hand, the overall low fraction of shared ride requests observed in the empirical trip records, even in the high-sharing regime, suggests that an additional societal change towards acceptance of shared mobility is required[43] to make the full theoretical potential of ride-sharing accessible[9,17]. A carefully designed incentive structure for ride-sharing users adapted to local user preferences is essential to drive this change and to avoid curbing user adoption or stimulating unintended collective states[44,45]. This is particularly relevant in the light of increasing demand as urbanization progresses[1]. In the broader context of macroscopic mode-choice behavior, e.g. between private car, ride-hailing or public transport, results and extensions of our model should be considered also from the perspective of rebound effects, such as more traffic induced by higher demand counteracting the benefits of ride-sharing. Nonetheless, the overall impact of more attractive ride-sharing on sustainability of urban transport is likely to be positive[12,27]. Overall, the approach introduced above can serve as a conceptual framework to work towards sustainable urban mobility by regulating and adapting incentives to promote ride-sharing in place of motorized individual transportation.

## Methods

**New York City ride-sharing data**. We analyzed trip data of more than 250 million transportation service requests delivered through high-volume For-Hire Vehicle

(HVFHV) service providers in New York City in 2019. The data is provided by New York City's Taxi & Limousine Commission (TLC)[33] and consists of origin and destination zone per request, pickup and dropoff times, as well as a shared request tag, denoting a request for a single or shared ride. We compute the average request rate across all data throughout 2019 taking 16 hours of demand per day as an approximate average.

For fixed origin-destination pairs we determine the sharing fraction as the ratio of the total number of shared ride requests and the total number of requests. Departure and destination zones represent the geospatial taxi zones defined by TLC[33]. However, we exclude zones without geographic decoding, nor name tag defined by TLC. For each individual analysis, we fix the origin zone and compute the fraction of shared rides to destination zones.

To illustrate the spatial sharing adoption (shown in Figs. 1 and 5c, e), we exclude destination zones where the total number of requests is less than 100 trips in the whole year 2019 to avoid excessive stochastic fluctuations (see Supplementary Note 1 and Supplementary Methods for details). We include these trips in the calculation of the average sharing fraction of the zone though they do not affect the averages due to their small number ($10^2$ compared to about $10^8$ trips in total).

**Chicago ride-sharing data**. We additionally analyzed more than 110 million trips delivered by three service providers in Chicago in 2019. The data is provided through the City of Chicago's Open Data Portal and contains, amongst others, information of trip origin, destination, pickup and dropoff times as well as information whether a shared ride has been authorized[46]. While information is available on whether a request was matched with another user, the flag denotes all consecutive trips where the vehicle was not empty, even if the passengers never shared part of their trip. We restrict ourselves to geospatial decoding of the city's 77 community areas, as well as trips leaving or entering the official city borders. In analogy to New York City, we compute the average request rate across all data for 2019 taking 16 hours of demand per day as an approximate average reference time and repeat the analysis explained for New York City.

**City topology**. For our ride-sharing model we construct a stylized city topology that combines star and ring topology[41]. Starting from a central origin node, rides can be requested to 12 destinations distributed equally across two rings of radius 1 (inner ring) and 2 (outer ring), as depicted in Fig. 3. The distances between neighboring nodes on the same branch are set to unity. Correspondingly, the distances between neighboring nodes are $\pi/3$ on the inner, and $2\pi/3$ on the outer ring.

**Ride-sharing adoption**. We compute the equilibrium state of ride-sharing adoption by evolving the adoption probabilities $\pi(d,t)$ following discrete-time replicator dynamics[47,48]

$$\pi(d, t + 1) = r(d, t)\, \pi(d, t), \tag{3}$$

where the reproduction rate $r(d, t)$ at destination $d$ and time $t$ is

$$r(d, t) = \frac{E[u_{\text{share}}(d, t)]}{E[u(d, t)]} = \frac{u_{\text{single}}(d) + E[\Delta u(d, t)]}{u_{\text{single}}(d) + \pi(d, t)E[\Delta u(d, t)]} \tag{4}$$

and $E[X]$ represents the expectation value of random variable $X$. Conceptually, each user observes their utility difference between single and shared rides over a number of rides (e.g. using the service for week) and then adjusts their strategy $\pi(d, t)$ for the next time step. Users thus effectively learn their optimal equilibrium strategies where they cannot increase their utility by changing their decisions.

We realize this process in the following way: We prepare the system in an initial state $\pi(d, 0) = 0.01$ of ride-sharing adoption for all destinations $d$, modeling the emergence of sharing. We fix the utility for a single ride $u_{\text{single}}(d) = 4$ (unless stated otherwise) to ensure positivity of Eqn. (4). The value of $u_{\text{single}}$ effectively controls the step size of the algorithm with $u_{\text{single}} \rightarrow \infty$ corresponding to the continuous time limit of the replicator equation. The choice of $u_{\text{single}}$ does not affect the equilibrium states ($\Delta u = 0$ or $\pi^* \in \{0, 1\}$) and only determines the speed of convergence (compare Supplementary Fig. 18). To evolve Eqn. (3), we numerically compute $E[u_{\text{share}}(d, t)] = E[u(d, t)|\text{share}]$ at each replicator time step $t$: We generate $n = 100$ samples of ride requests of size $S$ of which at least one goes to destination $d$ and requests a shared ride. The other $S - 1$ requests are drawn from a uniform destination distribution. Each of them realizes a sharing decision in line with the current probability distribution $\pi(d', t)$ at their respective destination $d'$ at time $t$. Shared ride requests are matched pairwise (see below). From these $n = 100$ game realizations, we compute the conditional expected utility of sharing. We repeat this procedure for all destinations $d$ and then update all probabilities $\pi(d, t)$ according to Eqn. (3).

Before performing measurements on the system's equilibrium observables, we evolve the system for 20000 replicator time steps, corresponding to two million game realizations per destination. We discard a transient of 19000 replicator time steps and quantify the degree of fluctuations per $\pi(d)$ around its mean value over time for the last 1000 time steps. If fluctuations do not exceed a threshold of two percentage points we consider the system equilibrated. Else, we continue to evolve the system for another 5000 replicator time steps, test whether the equilibrium

threshold is met, and potentially repeat the procedure. The average ride-sharing adoption $\langle \pi(d) \rangle$ over the last 1000 replicator time steps represents a proxy for the stationary solution $\pi^*(d)$ of Eqn. (3) and is plotted as the sharing fraction in Figs. 3 and 4. In Supplementary Fig. 19 we quantify the degree of fluctuations per parameter constellation in the phase diagram in Fig. 4a and demonstrate a high degree of equilibration, much better than the required threshold.

**Heterogeneous preferences**. Simulations for users with heterogeneous convenience preferences are carried out for fixed inconvenience parameters $\zeta_i$ for different user types. To determine the equilibrium ride-sharing adoption per user type, we repeat the equilibration procedure as explained in the previous paragraph, but the $S$ requests consist of randomly chosen user types with different inconvenience preferences. The probability to draw a user with preference $\zeta_i$ is given by the exogenous parameter $\Pr[\zeta_i]$ (see Supplementary Note 4 and Supplementary Fig. 15).

To produce Fig. 4c we fix $\epsilon = 0.2$, $\zeta_1 = 0.172$ and $\zeta_2 = 0.270$. The probabilities to draw $\zeta_1$ or $\zeta_2$ are $\Pr[\zeta_1]=0.4$ and $\Pr[\zeta_1]=0.6$, respectively. Other values yield qualitatively similar results (see Supplementary Note 4 for details). To compute the macroscopically observed combined contribution of shared ride requests from both user types (gray in Fig. 4c), we sum the number of shared ride requests from the two user types for given total demand $S$.

To study the approximate macroscopic ride-sharing dynamics of a real city we superimpose 600 origin zones with different local demand for rides and local differences in convenience preferences of users (compare Fig. 4d). Per origin we determine the local demand $S$ from an exponential distribution (see Supplementary Note 4 for details). Per origin, users may segment into three groups of convenience preference types $\zeta_i \in \{0.175, 0.225, 0.275\}$. The probabilities $\Pr[\zeta_i]$ govern the distribution of convenience types per origin. Note that the distribution also determines for how many people, on average, $\epsilon$ overcompensates potential inconvenience effects.

Across origins we fix the macroscopic average ratio of financial incentives to inconvenience at $E[\epsilon/\zeta] = 1.05$, hinting at a full-sharing state at the aggregated level. We draw the probabilities $(\Pr[\zeta_1], \Pr[\zeta_2], \Pr[\zeta_3])$ from a normal distribution with mean $E[\epsilon/\zeta] = 1.05$ and standard deviation $\sigma = 0.085$, fixing the local ratio of financial incentives to expected inconvenience parameters (see Supplementary Fig. 16).

**Matching**. Each request set of size $S$ decomposes into single and shared ride requests. We realize the optimal pairwise matching of requests as follows: For shared requests we construct a graph whose nodes correspond to requests and edges encode the distance savings potential of matching the two requests. To determine the distance savings potential we assume that, independent of single or shared ride, the provider has to return to the origin of the trip.

After constructing the shared request graph we employ the 'Blossom V' implementation of Edmond's Blossom algorithm to determine the maximum weight matching of highest distance savings potential[49,50]. The matching determines the routing and the realization of inconvenience and detour (see Supplementary Note 3 for more details). Since in the model all user requests are served, this matching strategy is consistent with a profit maximizing service provider.

## Data availability

The trip record dataset for New York City is available in the Taxi & Limousine Commission's (TLC) public repository https://www1.nyc.gov/site/tlc/about/tlc-trip-record-data.page as 'High Volume For-Hire Vehicle Trip Records'[33]. The trip record dataset for Chicago is available on Chicago's Open Data portal https://data.cityofchicago.org/Transportation/Transportation-Network-Providers-Trips/m6dm-c72p as 'Transportation Network Providers - Trips'[46]. The simulation datasets generated from the game theoretical model in the current study are available upon reasonable request to the authors.

## Code availability

Full details on the data analysis and game theoretic modeling are provided in the Supplementary Information. The simulation code is available in the public Github repository 'PhysicsOfMobility/ridesharing-incentives', https://doi.org/10.5281/zenodo.463050851.

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

## Acknowledgements

We thank the Network Science Group from the University of Cologne and Nora Molkenthin for helpful discussions and Christian Dethlefs for help with simulations. D.S. acknowledges support from the Studienstiftung des Deutschen Volkes. M.T. acknowledges support from the German Research Foundation (Deutsche Forschungsgemeinschaft, DFG) through the Center for Advancing Electronics Dresden (cfaed).

## Author contributions

D.S. initiated the research with help from M.S. and M.T. All authors designed the research and provided methods and analysis tools. D.S. collected and analyzed the empirical data with help from M.S., D.S. and M.S. designed and analyzed the game theoretic model. All authors contributed to interpreting the results and wrote the manuscript.

## Funding

## Competing interests

The authors declare no competing interests.
