## [Peer Review File · Nature Communications]

Reviewer #1 (Remarks to the Author):

- Summary:

- o Ridesharing (in contrast to more general ride-hailing) can substantially improve mobility, as it is efficient for on-demand vehicles to carry multiple people to common destinations
- o Modeling ridesharing as a game, the authors find two regimes: ride-sharing increasing with the rate of rides requested and ridesharing that is independent of the rate of rides requested
- o The authors claim that these results are also seen in two American cities, and therefore suggest that increasing ride-sharing incentives can substantially increase ridesharing

- What are the major claims of the paper?

- o Ridesharing can be modeled as a game
- o Simulations show two regimes in which ridesharing increase with the ride request rate or is constant
- o Comparisons are made with empirical data, and the authors claim that the data qualitatively agrees with the simulation.

- Are they novel and will they be of interest to others in the community and the wider field? If the conclusions are not original, it would be helpful if you could provide relevant references.

- o These results are novel, and if the questions I discuss below are addressed, they would be of great interest to the wider community due to the potential policy implications.

- Is the work convincing, and if not, what further evidence would be required to strengthen the conclusions?

- o As-is, I do not believe the results are convincing.
- o First, the policy implication of increasing benefits to ride-sharers leaves out the potential problem in which this would create an overall increase of ride-hailing users by increasing ride-sharers in a more realistic condition in which the number of ride-hailers is a dynamic variable that is affected by game theoretic motivations. Therefore, there might be an overall increase in car traffic. This would make the conclusions of the paper mute, if this counters the benefits of ridesharing.
- o I also have questions about the simplistic road/mobility network model. Road networks and mobility networks in real cities diverge substantially from this simplistic network. Do the qualitative results hold in more realistic networks?
 - i. For example, one can gather the road network from NYC or Chicago via Open Street Map shapefiles, and open these files with Python's NetworkX library to find the road network topology. The US Census also says where people commute within census tracts, thus we know where origins and destinations are, at least for work commuting (pre-COVID-19 anyway).
- o Most importantly, the empirical results as-is are not particularly convincing. Namely there is not an obvious difference between two regimes in empirical data. This is most clearly shown in Supplementary Fig. 3, which removes the dashed lines. As an example, nothing intrinsically suggests LGA and Crown Heights North belong to separate distributions, especially with the lack of data in the limit of many rides/minute. Instead the variance we see could be due to heteroskedasticity in the data and not because the data lies in a "positive slope" and "zero slope" regime (dashed lines in Fig. 5). Also note that Supplementary Fig. 5 shows datapoints not shown in main text. When we see these outliers, it becomes more clear that all data follows a roughly linear trend with wide (heteroskedastic) variance. I do not see a strong argument that any of the data follows a constant ride share regime (zero-slope dashed line).

- On a more subjective note, do you feel that the paper will influence thinking in the field?

- o If the results are proven to more convincingly reflect data, then yes.

- Please feel free to raise any further questions and concerns about the paper.

- o Major comments:

- ♣ To address the empirical questions I stated earlier, is there a way to directly measure the utility costs? This could help us fix the values of the dashed lines in the empirical data shown in Fig. 5 a,b. If there is a way to fix these dashed lines, and perhaps if we find better qualitative agreement with data, then these results will appear more convincing.

- While I am not fully convinced by the empirical data, the results are still quite interesting
- The ride matching, simple model, and game theoretic model are quite nice, but better empirical validation is in my opinion critical to make this a Nature Communications paper rather than, e.g., a Scientific Reports paper.
- It would be nice to provide a more direct comparison between model and data, e.g., does any data demonstrate a negative correlation between utility or demand and the number of miles users rideshare?
- ♣ How does noise in the system affect results? Humans are far from perfect, and the utility calculations people make are imperfect – there is no reason to expect that users know the dollars and cents cost and benefit of sharing vs not sharing a ride.
- o Minor comments:
 - ii. "Ridesharing" is a bit ambiguous, although I fully understand the authors have done their best to clarify the meaning of ridesharing – as opposed to ridehailing – in the first sentence of the abstract. Perhaps they could change "Ride-sharing constitutes a promising alternative to individual motorized transport..." with "Ride-sharing constitutes a promising alternative to individual motorized transport, such as ride-hailing,..." The ambiguity arises from a lot of literature that considers ridehailing and ridesharing as one and the same. Even Wikipedia says "The term "ridesharing" has been used by many international news sources, including The Washington Post,[2] CNN,[3] BBC News,[4] The New York Times,[5] the Associated Press,[6] and the Los Angeles Times.[7][8] Groups representing drivers, including Rideshare Drivers United[9] and The Rideshare Guy (Harry Campbell),[10] also use the term "rideshare", since "hailing" rideshare cars from the street is illegal. Usage is inconsistent, with the same publication or the same article sometimes using both "ridesharing" and "ridehailing".[11]"
 - iii. Benefit of discussing some SI figures, e.g., Fig. 2 and 10 in the SI. E.g., in Fig. 10 are these the same networks with the same origins as the main text? I believe so, but the next figure discusses a new origin, which makes SI Fig. 10 less clear than I think it should be.
- We would also be grateful if you could comment on the appropriateness and validity of any statistical analysis, as well the ability of a researcher to reproduce the work, given the level of detail provided.
- o The authors are only providing code upon request. I would recommend they post it in a public repository for better reproducibility

Reviewer #2 (Remarks to the Author):

This is a review report for manuscript entitled 'Incentive-driven discontinuous transition to high ride-sharing adoption' by Storch et al., which has been submitted for consideration in Nat. Comms. with manuscript number NCOMMS-20-32901.

The authors explore the complex interplay between ride sharing opportunities and collective decision-making via a game theoretical model where users have both positive incentives to share rides (e.g. ride discounts), as well as negative incentives to avoid sharing (detours, reduced privacy, etc.). The model shows two regimes of ride-sharing adoption: 1) When the financial incentive is large enough, sharing increases linearly with demand (high sharing regime); 2) When the negative incentives are relatively large, sharing saturates and becomes constant for increasing demand (low sharing regime). Interestingly, the modeled dynamics in the low sharing regime turns into an anti-coordination game where neighboring areas may have opposite sharing strategies, implying spatial heterogeneity in sharing patterns. To show the relevance of their modeling efforts, the authors analyze real-world ride-sharing data from NYC and Chicago, finding (some) qualitative evidence of both high- and low-sharing regimes in these cities.

The article is well-written and clear, and to the best of my understanding scientifically sound. The results of the authors are quite relevant, in the sense that their analysis of ride sharing not only considers the efficiency benefits of increased sharing at the systemic level, but explicitly takes into account the decisions of individuals (i.e. choosing comfort instead of price reductions) in order to

place limits on the applicability of ride-sharing as a tool of city-wide transport. In general, my only major issue is the somewhat shallow comparison between data and model (not going much beyond the distribution of locations based on sharing ratio of Fig. 5) plus a few technical remarks that I describe in detail below. I invite the authors to revise their manuscript based on these comments. After the revision, I believe the manuscript will have the quality and relevance necessary to ensure publication in Nat. Comms.

GENERAL COMMENTS

- The authors mention several times that 'the data suggests that even a moderate increase of financial incentives may strongly improve ride-sharing adoption in some areas currently in the low-sharing regime'. While this is clear in the model (i.e. if parameter values are close enough to the transition line in the phase space of Fig. 4), it is far less clear in the data. Indeed, the empirical sharing patterns in Fig. 5 suggest sharing regimes qualitatively similar to those of the model (a diagonal of high-sharing locations plus a few locations in the low sharing regime), but it is not clear to me how this translates to e.g. an estimated value of epsilon that tells us how far away the data is from the transition line in a fitted model. I suggest the authors to clarify this issue, especially since it relates to one of the most relevant consequences of their analysis in terms of the real-world applicability of the model. One option is just to tune down the tone of this conclusion, since it might be unwarranted. Another option (my preferred one, but more time-consuming) is to do something along the lines of data-driven modeling: take some properties of the real data (topology of the city network, distribution of ride requests, etc.), simulate the model on top, and try to at least qualitatively match the empirical distribution of shared trips, to get a hint on the parameter values (epsilon, etc.) that would correspond to NYC and Chicago, and then actually determine whether the city is close to a transition line and would thus benefit from a 'moderate increase of financial incentives'. Without a closer link between data analysis and the model dynamics, I feel that the conclusions drawn from the perceived two regimes in Fig. 5 cannot really extend to specific suggestions on how to increase shareability in real-world cities.

TECHNICAL COMMENTS

- Is there an empirically motivated reason to choose the value $u_{\text{single}} = 10$ for the single-ride utility? (beyond ensuring positivity of Eq. 5 as stated in Methods). Are there qualitative differences in the behavior of the replicator dynamics as a function of u_{single} ? (There's some mention of this around Eq. S5 in the SI where the authors mention that the infinite limit corresponds to a continuous replicator dynamics.) Perhaps it would be useful to mention somewhere that the regimes of adoption are robust to the choice of u_{single} , and also mention how u_{single} could be related to some measurable quantity (i.e. ride price).

- Figure 3:

The labels for strategies at the bottom of Fig. 3 b-e are somewhat unclear (inner share, outer share, etc.). I guess the associated incentive plots correspond to the incentive values averaged over subsets of modeled users in certain areas (inner, outer) with given chosen behaviour (share, single, etc.), right? Further, the 'mixed' strategy (in subplots d,e) corresponds to utility $u_{\text{share}} = 0$, right? Perhaps all of this should be written more clearly, at least in the caption. Also, in subplot b, why are the strategies 'inner share' and 'outer share' repeated two times?

- Figure 5:

Homogeneous vs. heterogeneous patterns of ride-sharing in the high-sharing / low-sharing regimes. Subplots c-f seem to show this dichotomy in the distribution of share-ride requests depending on what regime the location is on. (A very interesting find! Particularly when comparing locations with the same request rate s .) Still, just looking at the color pattern of subplots c-f doesn't convince me that e.g. LGA is more heterogeneous than East Village. Could you add some other quantitative measure of this spatial heterogeneity / lack thereof? Plus some measure that helps convince us of the statistical significance of the difference in heterogeneity between locations in both regimes? This find is clear in the idealized cities of Figs. 3-4 and the SI, but much less so in the data for NYC and Chicago.

- Another comment on the comparison of spatial homogeneity and heterogeneity between sharing

regimes: While comparing East Village and LGA in the NYC dataset makes sense since they have similar values of s , comparing Austin and O'Hare in the Chicago dataset is a bit misleading: could it happen that Austin is more homogeneous simply because there are less rides per unit of time (i.e. lower s)? (and again, even then, the pictures almost look similarly heterogeneous to me) A better measure of spatial heterogeneity might help clarify this issue.

MINOR COMMENTS

- In the references to the SI (text 'see Supplementary Information for details' throughout the manuscript), it would be good to have an explicit reference to the SI section the authors refer to, to make it easier for the reader to find the corresponding information.

- There's some mismatch of references to the main text in the SI and the main text itself. For example, in the caption to Fig. S4 there's a reference to Fig. 4 d in the main text (the subplots only go up to c). Another example are the references to Fig. 5 subplots in the main text (page 6), which seem to be mixed (i.e. referring to heterogeneities in Fig. 5 c, instead of d?). There might be some others I didn't catch, so I suggest a careful check of figure references, etc.

Reviewer #3 (Remarks to the Author):

The authors present a study of the dynamics of ride-sharing adoption in the context of on-demand transportation systems. The authors first model the problem at a theoretical level as an anti-coordination game, then apply the model to a simple synthetic urban network to uncover discontinuous transitions in the system dynamic. The most interesting part of the paper is the last, in which the authors show that those transitions can be observed -- at least to some extent -- also on large-scale ride-sharing data sets collected from NY and Chicago.

While I find the paper interesting and the model novel, I think the authors can further improve it under several respects.

First, the model is based on strong assumptions, that should at least be discussed. In particular, the model is based on the non-stated assumption that, when taking a decision on whether requesting a single or shared ride, the user has "perfect knowledge" of the expected probability of actually sharing the ride, not only a general sense, but a very detailed estimate based on the selected destination. I doubt that this is the case in practical scenarios. Notice that the probability of sharing the ride depends also on behavioral choices of other users, so in a sense the model assumes that users have a good sense of what all other users in the ride sharing market typically do. This is very rich knowledge that I believe an operator like Uber or Lyft might have based on millions of observations. However, I do not find realistic the assumption that a regular user has access to this knowledge as well. Would the model work also in the context of bounded user knowledge?

Authors assume that matching is done based on the maximum number of saved miles. However, I hardly believe that this would be the parameter used by a mobility operator, which would probably prefer to maximize the total number of riders served. In fact, in a highly competitive market like NYC/Chicago, operators might want to increase their market share, i.e., total number of served customers. In this context, that would translate in optimizing for maximum number of matched rides, rather than maximum number of saved miles.

In the data-based analysis, the authors filter out what they call "infrequent" destinations, using a threshold value of 100 trips. The authors mention that this is done to avoid fluctuations. While this is reasonable, I would like to see a robustness analysis on how the value of the threshold used for filtering might impact the results. One of the reasons for doing this is that, while most of the trends highlighted by the authors in the model and simulations can be observed on real data, the "bifurcation" trend -- which is actually one of the main findings of the model analysis -- can only be hinted to by looking at the data. Looking at Figure 5, the trend is only barely visible in NY, with very few dots concentrating along the horizontal orange line. Even less in Chicago, especially

looking at Supplementary Figure 5, where the few dots (no more than 5) close to the orange line seems to be outliers of a general sublinear trend (see inset in the same Figure), rather than delineating a clearly separate trend. At the very least, the authors should show that these "hints" to bifurcation observed in the big data analysis are preserved and not an artifact of choosing a specific value of trips for filtering out infrequent destinations.

It would be also useful to add a plot showing the evolution of $\pi(d,t)$ with epochs, to assess that the system actually reaches an equilibrium, and the speed of convergence to equilibrium.

The authors should clarify whether the "shared ride" flag in the NY and Chicago data set refers to the fact that the ride sharing option has been selected, or otherwise to rides that have actually been shared. My understanding is that the first option holds, at least for Chicago. If so, is there any information in the data sets on which of the requests have actually been performed in ride sharing mode? If available, this information would be very useful to attempt to further validate the model, i.e., comparing the ratio of actually shared rides vs total shared requested rides.

In the SI, page 11, there is a reference to Figure 11, which should be Figure 10 instead.

Replies to the comments of the reviewers

on the manuscript

Incentive-driven transition to high ride-sharing adoption

by David-Maximilian Storch, Marc Timme and Malte Schröder

GENERAL REPLIES

We thank all three reviewers for their thorough and detailed comments on the manuscript and their comprehensive and positive evaluation of our work. The reviewers judge the study as "novel", "of great interest to the wider community" and consider the manuscript "well written". The reviewers remark on a number of additional aspects relevant to ride sharing adoption, including rebound effects and the choice of incentives, requesting further discussion on these points. Additionally, the reviewers ask us to provide more details about the empirical data to strengthen support for our modeling approach; one reviewer specifically asks us to discuss the possible heteroskedasticity observed in data in relation to our model findings.

The main aim of the game-theoretic model presented and analyzed in the manuscript is to capture the fundamental incentives of customers for and against adopting ride-sharing services rather than individual-service ride-hailing. As such, the model (i) provides a novel perspective on the dynamics of ride-sharing services beyond existing work focusing on the viewpoint of the service provider, (ii) offers one possible, consistent explanation of the qualitative features observed in ride-sharing adoption in New York City and Chicago, and most importantly (iii) enables mechanistic insights about the consequences of which factors may enhance or inhibit adoption – and how. Our approach goes beyond individual empirical studies, thereby providing a conceptual basis for future studies and potential policy making.

In particular, motivated by empirical findings, we observe and explain distinct phases of ride-sharing adoption in the simplified model and uncover a discontinuous transition between them in the high-demand limit. While we expect this transition to be less sharp in settings with a finite number of players and heterogeneous demand, as observed in the empirical data, our main conclusion holds more broadly: a change in (financial) incentives may have a disproportionately large impact on the ride-sharing adoption close to such a transition. In this sense, we would expect a change of the incentives to strongly (but not

discontinuously) affect ride-sharing adoption in some of the zones in the low-sharing regime. We have removed the word 'discontinuous' from the title to avoid possible confusion that the predicted transition is always discontinuous, even in heterogeneous and finite-size systems.

To conclusively identify the mechanisms underlying ride-sharing adoption in any given real scenario, one would need to conduct experiments in which incentives (such as pricing) are systematically varied given other relevant factors staying roughly constant. While such a study is clearly beyond the scope of our study (which introduces the modeling and analysis approach in the first place), indications of such a mechanism may be directly observable in the response of customers when, for example, ride-sharing services change their pricing.

We have carefully revised the manuscript and the Supplementary Information accordingly, highlighting the core aspects of our results. In particular, we provide a more detailed discussion of our model and the fundamental incentives, including a number of additional references and additional empirical observations supporting our choice of incentives. Additionally, we present more details on the simulations and complement the original simulation results by new ones investigating settings that more realistically reflect real city ride-sharing dynamics, e.g. through a more diverse user base with heterogeneous preferences. These new results are consistent with the empirical observations from New York City and Chicago.

We have furthermore revised the discussion to include key points raised by the reviewers and to more specifically highlight which aspects of the empirical data are explained by the model, at the same time also discussing potential alternative explanations and open questions. We address all comments in detail in the point-by-point responses below.

REPLIES TO THE COMMENTS OF REVIEWER 1

Reviewer 1 considers the manuscript "interesting", "novel" and finds that a revised work may be "of great interest to the wider community" with "potential policy implications" and may "influence thinking in the field". The reviewer remarks on several additional effects that may be relevant in the context of ride-sharing adoption, including rebound effects and human imperfection in their decision making. The reviewer requests additional details on the empirical validation before recommending publication.

Reviewer comment:

Ridesharing (in contrast to more general ride-hailing) can substantially improve mobility, as it is efficient for on-demand vehicles to carry multiple people to common destinations. Modeling ridesharing as a game, the authors find two regimes: ride-sharing increasing with the rate of rides requested and ridesharing that is independent of the rate of rides requested. The authors claim that these results are also seen in two American cities, and therefore suggest that increasing ride-sharing incentives can substantially increase ridesharing.

Major claims:

- *Ridesharing can be modeled as a game*
- *Simulations show two regimes in which ridesharing increase with the ride request rate or is constant*
- *Comparisons are made with empirical data, and the authors claim that the data qualitatively agrees with the simulation*

These results are novel, and if the questions I discuss below are addressed, they would be of great interest to the wider community due to the potential policy implications.

Authors' response: We thank the reviewer for their concise summary and positive judgement of our work in the light of its great interest to the wider community and its potential policy implications.

Reviewer comment:

As-is, I do not believe the results are convincing. First, the policy implication of increasing benefits to ride-sharers leaves out the potential problem in which this would create an overall increase of ride-hailing users by increasing ride-sharers in a more realistic condition in which the number of ride-hailers is a dynamic variable that is affected by game theoretic motivations. Therefore, there might be an overall increase in car traffic. This would make the conclusions of the paper mute, if this counters the benefits of ridesharing.

Authors' response:

We thank the reviewer for raising this important point.

The rebound effects mentioned by the reviewer occur on a macroscopic mode-choice level for decisions between, for example, private cars, ride-sharing, public transit, or cycling: relatively more attractive ride-sharing compared to the other modes (e.g. through higher financial incentives) is expected to induce higher demand for ride-sharing coming from other modes (or from trips that otherwise would not have taken place).

Related research suggests that these rebound effects may be much more relevant to the overall traffic (number of cars on the road) than for the (environmental) benefits of ride-sharing [López et al., PNAS 111 (51) E5488 (2014), Santi et al., PNAS 111 (51) E5489 (2014)]. For example, Morris et al. (2019) conclude that "[ride-sharing] may compete nearly as much with private vehicle travel and transit travel." [Morris et al., *Assessing the Experience of Providers and Users of Transportation Network Company Ridesharing Services*, Tech. Report, (grant no. 69A3551747117) (2019), p. 124f.]. While substitution of public transit trips is undesirable, replacing private vehicle trips by ride-sharing services may have positive environmental effects. Moreover, Jenn (2020) shows that electrified ride-hailing vehicles have a three times higher emission reduction potential than privately owned electric vehicles [Jenn, *Nat Energy* 5, 520–525 (2020)] and electrified pooled rides may further enhance this ratio [Anair et al., *Ride-Hailing's Climate Risks: Steering a Growing Industry toward a Clean Transportation Future*. Cambridge, MA: Union of Concerned Scientists (2020) <https://www.ucsusa.org/resources/ride-hailing-climate-risks>, accessed on 2021/01/26].

In our model, we focus on the choice between single and shared rides of ride-hailing users. We specifically consider the state of ride-sharing adoption under given equilibrium

conditions, i.e. for a fixed number of users, fixed preferences and discounts. While rebound effects may increase the number of users S , they do not change the qualitatively different states of equilibrium adoption discussed in the manuscript. Such effects are covered by our analysis by considering that a change of the financial incentives may depend on a change of the number of users when moving in the phase diagram of ride-sharing adoption (Fig. 4a). Importantly, these rebound effects do not move the system from the high- to the low-sharing regime.

In the revised manuscript, we have added a brief discussion of rebound effects and their implications both in the model description as well as on the macroscopic mode-choice level.

Reviewer comment:

I also have questions about the simplistic road/mobility network model. Road networks and mobility networks in real cities diverge substantially from this simplistic network. Do the qualitative results hold in more realistic networks? i. For example, one can gather the road network from NYC or Chicago via Open Street Map shape files, and open these files with Python's NetworkX library to find the road network topology. The US Census also says where people commute within census tracts, thus we know where origins and destinations are, at least for work commuting (pre-COVID-19 anyway).

Authors' response:

We thank the reviewer for their intriguing suggestion.

Our model consists of a set of coupled replicator equations that capture the interdependencies of collective ride-sharing decisions, street network topology and origin-destination demand distribution. For three reasons extended simulations on a realistic city scale are unfortunately not computationally feasible without access to super-computing resources over extended periods of months to years: (i) intrinsic algorithmic complexity and corresponding simulation runtimes, (ii) increasing equilibration times of the replicator dynamics, (iii) multiple simulation runs for parameter scans and robustness analysis.

- (i) *Intrinsic algorithmic runtime hurdles:* Real-city scale simulations may have hundreds of different destinations and locally hundreds of simultaneous shared ride requests, as observed at some locations in the empirical New York City or Chicago trip records.

Together with the intrinsic cubic scaling of the perfect matching algorithm in the number of shared ride requests, simulations at this scale quickly become intractable even on super-computers.

- (ii) *Increasing equilibration times*: On the one hand, the number of required Monte Carlo samples to compute players' utility increases strongly with the number of players and different destinations. We have to accurately sample all configurations of different ride-sharing decisions and destinations of the S players according to the given origin-destination distribution for all player types. On the other hand, heterogeneity of the average distances and origin-destination distribution in real cities translates into heterogeneous equilibration timescales per player type, thus multiplying the total equilibration as all players' decisions may have to (re)equilibrate multiple times when slower players change their decisions. Overall, we expect real-city scale simulations to significantly increase the timescales and game repetitions required to equilibrate the replicator dynamics (see also replies below).
- (iii) *Multiple simulation runs*: To draw robust conclusions from real-city simulations and to reproduce a phase diagram like the one shown in Fig. 4 we would need to systematically repeat the simulation runs for many parameters (e.g. request rate) to unveil the phase transition in the ride-sharing adoption. Such variations would further multiply the increases in runtime sketched above.

For this purpose we have *deliberately* designed the model for the incentive structure and for the underlying network to be as simple as possible. Due to its simplicity, the model is capable of revealing mechanistic insights into the dynamics and interactions underlying ride-sharing adoption, as we can analytically understand these mechanisms. While more complex, heterogeneous, high-dimensional real world systems may introduce additional effects, the fundamental mechanisms observed in our model persist, thus providing a baseline mechanism for explaining different observations.

The above limitations of possible simulation scenarios notwithstanding, our additional simulations in various alternative settings and an exact mathematical proof pinning down the core mechanism of the transition both provide strong additional evidence for the robustness of our main result.

To further demonstrate the robustness of our model results we systematically vary three major sources of heterogeneity known from real-city settings: (i) We introduce heterogeneous origin-destination demand structures in our simplified topology and assess its impact on the existence of different regimes of ride-sharing adoption. The results are reported in Supplementary Note 4. While the origin-destination demand structure guides the formation of spatial patterns of ride-sharing adoption, it does not impact the existence of disparate adoption regimes. (ii) We model heterogeneous street network topologies by varying the position of the trip origin, also reported in Supplementary Note 4, thereby breaking the radial symmetry of the city layout. While such topological changes may alter spatial patterns of ride-sharing adoption, low and high ride-sharing adoption regimes again persist. (iii) We introduce heterogeneous player types by varying the relative importance of inconvenience across the players. While the transition by construction becomes blurred out, the individual player types still feature a discontinuous change in their adoption behavior as predicted already by the homogeneous model (compare Fig. 4 in the revised version of the main manuscript and Supplementary Figures 15-16). In Supplementary Note 3 we provide an exact mathematical derivation of the discontinuous transition between the two adoption regimes with homogeneous players, irrespective of street network topology, or origin-destination demand distribution.

We have furthermore added the simulation results for heterogeneous customer types identified by different convenience preferences to the revised manuscript (see Fig. 4c,d) and included an extended discussion about the decentrally positioned origin and the mathematical derivation to further illustrate the robustness of our results.

Reviewer comment:

Most importantly, the empirical results as-is are not particularly convincing. Namely there is not an obvious difference between two regimes in empirical data. This is most clearly shown in Supplementary Fig. 3, which removes the dashed lines. As an example, nothing intrinsically suggests LGA and Crown Heights North belong to separate distributions, especially with the lack of data in the limit of many rides/minute. Instead the variance we see could be due to heteroskedasticity in the data and not because the data lies in a "positive slope" and "zero slope" regime (dashed lines in Fig. 5). Also note that Supplementary Fig. 5 shows datapoints not shown in main text. When we see these outliers, it becomes more clear that

all data follows a roughly linear trend with wide (heteroskedastic) variance. I do not see a strong argument that any of the data follows a constant ride share regime (zero-slope dashed line).

Authors' response:

We thank the reviewer for inquiring about the empirical data.

We agree with the reviewer that the empirical data do not show as clear a distinction between the different regimes of ride-sharing adoption as observed for our model simulations with homogeneous user preferences and other explanations besides the incentive structure in our model may in principle be responsible for some of the observed features. Finite request rate in the empirical data as well as diverse, heterogeneous preferences of the ride-sharing users, among other factors, may offer explanations for these observations; in any case, our model predictions are in qualitative agreement with observations and consistent with different regimes of ride-sharing adoption (low to high) found in data.

- *Finite request rate:* While the transition between the high and low sharing regime is discontinuous in the high-demand limit ($S \rightarrow \infty$, i.e. in an infinitely large system as required to study phase transitions), all intermediate points exist for finite demand. In particular, in settings with a finite number of players, as the ratio of financial incentives to inconvenience ϵ/ζ crosses the critical value $\epsilon_c/\zeta_c = 1$, the sharing adoption changes continuously (but increasingly quickly) from the high to the low sharing regime (compare phase diagram in Fig. 4a and Supplementary Fig. 18). Refined and more detailed simulations of our model reveal a broader partial sharing phase (stable percentage of the population adopting ride-sharing, light blue in Fig. 4a) separating the low and high-sharing regime. This regime exhibits a linear increase of the number of ride-sharing users with a smaller constant fraction $S_{\text{Share}} = \alpha S$ with $\alpha < 1$ (see Fig. 4 in the revised manuscript). This partial sharing phase then transitions into the horizontal low sharing regime as S becomes large. These results are unchanged from the original version, but the transition was blurred out due to slow convergence to the steady state close to the transition (see our reply and Fig. 6 below).

Transitioning from high to low-sharing states via the partial sharing regime may be observed as a flattening of slope, or general sublinear trend in empirical data.

- *Heterogeneity*: In the empirical trip data, for example for Chicago, the number of realizations of sharing decisions may differ significantly across zones. Supplementary Fig. 6 shows large zones from the city center with extremely high demand of up to 50 requests per minute (Near North Side, Loop, Near West Side, Lake View) with an intermediate sharing fraction. It is likely that these high-demand zones summarize the ride-sharing behavior of a wide range of the population with heterogeneous ride preferences (e.g. convenience or privacy requirements).

To separate incentive-driven determinants of ride-sharing adoption from those stemming from heterogeneity or other factors, we focus on homogeneous types of user preferences in our initial model setup. In homogeneous settings, the different adoption regimes develop as described and lead to a discontinuous phase transition in the high demand limit.

In response to the issue of heteroskedasticity, we have now further analyzed the ride-sharing adoption on an aggregate level across multiple origins with different request rates and preference distributions. The results show (as expected given the reasons above) a clear heteroskedastic cone for low demand and a saturating, on average sub-linear trend for high demand (see Fig. 4d in the revised manuscript). This observation is qualitatively consistent with the trip records from Chicago and New York City.

We remark that the different states of ride-sharing adoption from the homogeneous configuration robustly persist in the heterogeneous setting, but are modulated by the distribution of preferences.

The observation of heteroskedastic variance is important to put observations of observables (a random variable) into context. However, often the root cause of dispersive variance in a random variable remains unclear. In combination, our new simulations and extended modelling offer one possible and consistent mechanism underlying the empirically observed heteroskedastic variation in the ride-sharing adoption, and the qualitative divergence observed in the data.

Importantly, our observation of a sudden transition between different adoption regimes remains valid for individual user types in heterogeneous settings. Thus, while we do not expect to observe a discontinuous transition in the empirical data, we still expect a change of the incentives to strongly (but not discontinuously) affect ride-sharing adoption in some

of the zones closer to the low sharing regime.

In our revisions, we have also further clarified the presentation of the results, in particular regarding the empirical observations, included our new simulation results on heterogeneous user preferences and now more carefully discuss the implications of our model.

Reviewer comment:

On a more subjective note, do you feel that the paper will influence thinking in the field? –
If the results are proven to more convincingly reflect data, then yes.

Authors' response:

We thank the reviewer for their positive judgement of the potential impact of the research on the thinking in its field.

Reviewer comment:

To address the empirical questions I stated earlier, is there a way to directly measure the utility costs? This could help us fix the values of the dashed lines in the empirical data shown in Fig. 5 a,b. If there is a way to fix these dashed lines, and perhaps if we find better qualitative agreement with data, then these results will appear more convincing.

Authors' response:

We thank the reviewer for their intriguing suggestion.

We agree with the reviewer that direct measurement of utility functions is an important prerequisite for quantitative predictions about critical financial incentives and demand-driven transitions between regimes of ride-sharing adoption for specific real cities, e.g. to help service providers or urban planners to refine local ride-sharing service offerings.

Our stylized model, however, is qualitative in its nature and analytically tractable for simple settings, intended to provide mechanistic insight into the collective interplay of ride-sharers. Hence, the model is not intentionally designed to predict city-specific critical parameters quantitatively, or to accurately fit empirical data in any city or region, e.g. due to its reductionist approach to limiting the number of heterogeneities or simplified assumptions about players' utility functions. While it may be theoretically possible to fit heterogeneous preference distributions in simulations with a realistic street network and demand distri-

bution, this approach seems computationally infeasible (compare our reply on extended simulations on realistic city-scale for New York City or Chicago).

It is, however, possible to provide additional indirect empirical evidence for our qualitative results using available census data and additional data on the ride-sharing trips. These indirect observations strengthen our modeling approach, confirming results of direct social science questionnaires that financial incentives, detour and convenience preferences are dominant contributions to a person’s utility in ride-sharing (see also the next reply for a more detailed discussion of this literature). For example, from the available trip records in Chicago, we find a strong negative correlation between the adoption of ride-sharing and the tipping behavior (tip as a fraction of the total fare, which is consistently smaller than in single rides of the same distance. Taking the tipping behavior as an indication of customer satisfaction [Bharat Chandar, B. et al., *The Drivers of Social Preferences: Evidence from a Nationwide Tipping Field Experiment*, Natural Field Experiments 00680, (2019)], this correlation may suggest a disutility of sharing a ride (termed *inconvenience* in our model).

FIG. 1. Tipping behavior in Chicago. (Left) Tipping is negatively correlated with the sharing fraction per origin zone. (Right) Across trip types, shared ride users tip less than single ride users.

A similar correlation (though much less strict) exists between the per capita income in a zone and the ride-sharing adoption in Chicago. These results suggest a relatively high importance of financial incentives, consistent with the observation of a more pronounced low sharing regime. In New York City, where the per capita income extends over a much broader range, per capita income and sharing fraction show no distinct relationship.

The distance dependence of ride-sharing adoption is more complicated as it is modified by the trip duration, which changes significantly over the course of a day (e.g. due to

FIG. 2. Adoption of ride-sharing services as a function of per capita income. (Left) In Chicago sharing fraction and income are negatively correlation. (Right) In New York City the sharing fraction is largely independent of income for per capita incomes, in particular for incomes well beyond fifty thousand US dollars [2017 inflation adjusted dollars according to census data]. Zero income zones include largely uninhabitated areas such as airports, Ellis Island, Jamaica Bay, public parks etc.

congestion). Nonetheless, we find that shared rides show a significantly broader distribution of trip duration (higher uncertainty) and a higher average duration when compared to single rides of the same distance in both New York City and Chicago (Fig. 3).

FIG. 3. Shared rides take longer on average and feature a higher variance, likely due to potential detours. Results are consistent for New York City [mean trip duration for single ride with distance between 9.75-10.25 km: 26.4 mins, and 31.5 mins for corresponding shared ride] and Chicago [mean trip duration for single ride with distance between 9.75-10.25 km: 20.1 mins, for corresponding share ride: 23.0 mins].

We have included these additional observations in the revised Supplementary Information and added a brief discussion, together with more detailed references supporting the choice of the incentives for the model (see next reply), in the revised manuscript.

Reviewer comment:

While I am not fully convinced by the empirical data, the results are still quite interesting. The ride matching, simple model, and game theoretic model are quite nice, but better empirical validation is in my opinion critical to make this a Nature Communications paper rather than, e.g., a Scientific Reports paper. It would be nice to provide a more direct comparison between model and data, e.g., does any data demonstrate a negative correlation between utility or demand and the number of miles users rideshare?

Authors' response:

We thank the reviewer for their positive judgement of our work and their suggestions for potential additional empirical evidence to more rigorously underpin the findings.

As detailed already above, we have now included a range of supporting observations and complementary findings into the revised manuscript and the Supplementary Information. In addition, empirical studies suggest that there exist strong negative correlations between the utility of ride-sharing and the experience of sharing the vehicle with other individuals [Morris et al., *Assessing the Experience of Providers and Users of Transportation Network Company Ridesharing Services*, Tech. Report, (grant no. 69A3551747117) (2019), Sarriera et al., *Transp. Res. Rec.* 2605(1), 109–117 (2017), Lippke et al., *Public acceptance and adoption of shared-ride services in the ride-hailing industry*, University of Michigan (2020)]. These studies suggest that the disutility of sharing a ride typically increases with the distance spent together in the vehicle or, for parts of the population, is a constant term. Additionally, the uncertainty about potential detours and the total trip duration is a major disutility consistently reported by ride-sharing users in the above mentioned literature contributions (compare also Fig. 3 in this reviewer reply for empirical trip duration distributions for shared and non-shared rides).

Morris et al. (2019) investigate rider and driver attitudes towards ride-sharing services in the US. In a survey with more than 1100 participants the authors study the perceived disutility of a shared ride compared to a single ride priced at 25 USD and lasting 20 minutes,

by examining participants' maximum willingness to pay for different levels of delay/service quality, very similar to the scenario modeled in our manuscript. Across three user groups – currently active users of ride-sharing services, former ride-sharers but now active solo riders, and thirdly, active solo riders who have never used the sharing service before – the authors find a linear negative correlation between the maximum willingness to pay for a shared ride and the maximum extra ride time compared to the single ride.

In a qualitative assessment of the ride-sharing experience, Morris et al. (2019, pp. 112-114) find consistent user reports about higher utility from shared rides without successful matching, as emphasized by the following quotations:

- *"Every time I take a lyft line and it ends up being just me the entire ride I feel like a genius"*
- *"I don't necessarily believe in luck, but if I did I would define it as 'taking a Lyft Line from Lakeview to River North without picking up any additional passengers.' YEP. THIS HAPPENED TO ME JUST NOW."*
- *"Whenever I order an UberPOOL I feel poor but if nobody joins during my ride, I feel très riche"*
- *"I took an UberPool from Williamsburg all the way to Hamilton Heights by myself! Which means I pretty much won the UberPool jackpot"*
- *"I hate when I get a Lyft line and the other passengers be so damn loud on their phone"*

In their sentiment analysis of more than 2000 Twitter messages on user experience during shared rides, they find approximately three times more negative vocalization (32.4%) about the service than positive (9.2%). The experienced positive or negative social interaction with matched other passengers is the dominant topic (59.3% of all messages) while sentiment about travel time (4.4%), or vehicle routing (6.0%) is expressed less often. This observation is consistent with Morales Sarriera et al. (2017) who find that, while both positive as well as negative social experiences exist during shared rides, the possibility of having a negative social interaction during a shared ride is a stronger disincentive to ride-sharing than it might be perceived positively to a potential user.

Based on a survey conducted with 1609 ride-hailing and -sharing users, as well as a focus group interview in Detroit, Michigan, Lippke et al. (2020) provide further evidence for

the need of strong financial incentives to encourage ride-sharing in the light of potential inconvenience and detours. The authors find significantly less willingness to pay for shared rides compared to single rides, decreasing in the number of additional passengers in the vehicle. Users reported insufficient financial discounts as a main reason for not adopting the sharing service.

In summary, the empirical literature on ride-sharing preferences suggests a utility of ride-sharing governed by (i) financial discounts, (ii) travel time uncertainties and detours, as well as (iii) inconveniences [Refs. 17-21, Morris et al. (2019, Table 9)], motivating and supporting our modeling choice of incentives and disincentives.

In the revised manuscript, we have added a discussion of this literature together with a more detailed and careful description of the empirical observations in combination with the additional simulations to illustrate the consistency of our model and the observed ride-sharing adoption (see above replies).

Reviewer comment:

How does noise in the system affect results? Humans are far from perfect, and the utility calculations people make are imperfect – there is no reason to expect that users know the dollars and cents cost and benefit of sharing vs not sharing a ride.

Authors' response:

We thank the reviewer for pointing out that noise, reflecting imperfect information or not fully rational human decision-making, might impact the robustness of our model findings and should be tested for.

As presented in the Supplement and briefly commented on in the revised manuscript, to consider imperfect decision-making and individual differences in perceived utility among riders we have conducted additional simulations where we add noise to the realized utility of shared rides. Taking $X \sim \mathcal{N}(0, \sigma^2)$ to be a normal random variable with mean zero and variance σ^2 , we realize the deterministic utility increment Δu as described in the manuscript and multiply it by a factor $(1 + x_{i,n,t})$ where $x_{i,n,t}$ denotes the realization of random variable X for rider i in game n at time t . The standard deviation σ acts as a control parameter for how different an individual might perceive the utility of a shared ride, or how strong external stochastic influences are. The ride-sharing adoption evolves based on the estimated

expected utility from 100 realizations (e.g. the experience of a group of similar users over a week). The users thus naturally average out large variances but do not arrive at a perfect estimate of their expected utility.

FIG. 4. Robustness of model results under utility fluctuations, reflecting imperfect information and heterogeneous utility perception.

For different noise amplitudes we consistently reproduce the average adoption behavior without noise (compare Fig. 4), demonstrating the robustness of our findings also for (i) strongly heterogeneous perception in the benefit of shared rides in the population and (ii) for stochastic service quality such as travel time. Please see also our replies to reviewer 2 regarding imperfect knowledge and the assumptions underlying the overall learning dynamics.

Close to transition points (e.g. at the transition to the high-sharing adoption regime), when the utility differences between different states are small, the impact of noise will be larger, yet the qualitative results and average sharing decisions remain robust.

We have added this discussion to Supplementary Note 4 and included a short mention in the revised manuscript.

Reviewer comment:

“Ridesharing” is a bit ambiguous, although I fully understand the authors have done their best to clarify the meaning of ridesharing – as opposed to ridehailing – in the first sentence

of the abstract. Perhaps they could change “Ride-sharing constitutes a promising alternative to individual motorized transport...” with “Ride-sharing constitutes a promising alternative to individual motorized transport, such as ride-hailing...” The ambiguity arises from a lot of literature that considers ridehailing and ridesharing as one and the same. Even Wikipedia says “The term "ridesharing" has been used by many international news sources, including The Washington Post,[2] CNN,[3] BBC News,[4] The New York Times,[5] the Associated Press,[6] and the Los Angeles Times.[7][8] Groups representing drivers, including Rideshare Drivers United[9] and The Rideshare Guy (Harry Campbell),[10] also use the term "rideshare", since "hailing" rideshare cars from the street is illegal. Usage is inconsistent, with the same publication or the same article sometimes using both "ridesharing" and "ridehailing".[11]”

Authors’ response:

We thank the reviewer for pointing out this ambiguity. Different publications (scientific and non-scientific) across different countries indeed use the terms ‘ride-hailing’ and ‘ride-sharing’ as well as sometimes ‘ride-pooling’ differently. These terms are often used to describe similar and sometimes identical concepts. We decided to follow the usage of ‘ride-sharing’ established in the literature of similar modeling studies [Santi et al., PNAS 111, 13290–13294 (2014), Agatz et al., EJOR 223, 295–303 (2012), Tachet et al., Sci. Rep. 7, 42868 (2017), Alonso-Mora et al., PNAS 114, 462–467 (2017)].

To avoid any confusion, we have amended the description in the abstract and the introduction as suggested by the reviewer and clearly defined the meaning of ‘ride-sharing’ and the difference to ‘ride-hailing’ in the revised manuscript.

Reviewer comment:

Benefit of discussing some SI figures, e.g., Fig. 2 and 10 in the SI. E.g., in Fig. 10 are these the same networks with the same origins as the main text? I believe so, but the next figure discusses a new origin, which makes SI Fig. 10 less clear than I think it should be.

Authors’ response:

We thank the reviewer for raising this point.

Supplementary Figure 2 continues the discussion of the empirical ride-sharing data from

NYC (including Supplementary Figs. 1-5). Supplementary Figure 10 to 16 refer to the same stylized city network shown in the main manuscript (central origin) with different demand distributions.

In our revisions, we have clarified the setting in the paragraph discussing these figures and structured the revised Supplementary Information to better convey the information on the simulation and data setting.

Reviewer comment:

The authors are only providing code upon request. I would recommend they post it in a public repository for better reproducibility.

Authors' response:

We thank the reviewer for encouraging us to make our simulation code publicly available in a public repository to ease reproducibility of our findings. The code is now publicly accessible in repository <https://github.com/Network-Dynamics/ridesharing-incentives>.

We note that the 'Blossom V' perfect matching algorithm embedded as a library in our code [Kolmogorov, Math. Prog. Comp. 1, 43–67 (2009)] is subject to license restrictions. The public repository for our code contains a reference and instructions on how to install the library for research purposes.

We have updated the code availability statement and provided a reference to the public repository of the code in the revised version of the manuscript.

REPLIES TO THE COMMENTS OF REVIEWER 2

Reviewer 2 considers the manuscript “well-written”, regards the study as “scientifically sound”, praises the results “quite relevant”, and highlights our discovery of disparate regimes of ride-sharing adoption “a very interesting find!”. The reviewer requests a few clarifications on technical details on the model and simulations as well as a more detailed discussion of the empirical evidence before recommending publication in Nature Communications.

Reviewer comment:

This is a review report for manuscript entitled ‘Incentive-driven discontinuous transition to high ride-sharing adoption’ by Storch et al., which has been submitted for consideration in Nat. Comms. with manuscript number NCOMMS-20-32901.

The authors explore the complex interplay between ride sharing opportunities and collective decision-making via a game theoretical model where users have both positive incentives to share rides (e.g. ride discounts), as well as negative incentives to avoid sharing (detours, reduced privacy, etc.). The model shows two regimes of ride-sharing adoption: 1) When the financial incentive is large enough, sharing increases linearly with demand (high sharing regime); 2) When the negative incentives are relatively large, sharing saturates and becomes constant for increasing demand (low sharing regime). Interestingly, the modeled dynamics in the low sharing regime turns into an anti-coordination game where neighboring areas may have opposite sharing strategies, implying spatial heterogeneity in sharing patterns. To show the relevance of their modeling efforts, the authors analyze real-world ride-sharing data from NYC and Chicago, finding (some) qualitative evidence of both high- and low-sharing regimes in these cities.

Authors’ response:

We thank the reviewer for this succinct summary.

Reviewer comment:

The article is well-written and clear, and to the best of my understanding scientifically sound. The results of the authors are quite relevant, in the sense that their analysis of ride sharing not only considers the efficiency benefits of increased sharing at the systemic level,

but explicitly takes into account the decisions of individuals (i.e. choosing comfort instead of price reductions) in order to place limits on the applicability of ride-sharing as a tool of city-wide transport.

Authors' response:

We thank the reviewer for their positive evaluation of the manuscript.

Reviewer comment:

In general, my only major issue is the somewhat shallow comparison between data and model (not going much beyond the distribution of locations based on sharing ratio of Fig. 5) plus a few technical remarks that I describe in detail below. I invite the authors to revise their manuscript based on these comments. After the revision, I believe the manuscript will have the quality and relevance necessary to ensure publication in Nat. Comms.

Authors' response:

We thank the reviewer for their suggestions. In our response to reviewer 1 we have touched on many of the comments already, which we will briefly summarize in our detailed responses below. In particular, we have added a more detailed analysis of the empirical data as well as of relevant supporting evidence for our modeling approach and additional details highlighting the robustness of our simulations in the revised manuscript and Supplementary Information.

Reviewer comment:

The authors mention several times that 'the data suggests that even a moderate increase of financial incentives may strongly improve ride-sharing adoption in some areas currently in the low-sharing regime'. While this is clear in the model (i.e. if parameter values are close enough to the transition line in the phase space of Fig. 4), it is far less clear in the data. Indeed, the empirical sharing patterns in Fig. 5 suggest sharing regimes qualitatively similar to those of the model (a diagonal of high-sharing locations plus a few locations in the low sharing regime), but it is not clear to me how this translates to e.g. an estimated value of epsilon that tells us how far away the data is from the transition line in a fitted model. I suggest the authors to clarify this issue, especially since it relates to one of the most

relevant consequences of their analysis in terms of the real-world applicability of the model. One option is just to tune down the tone of this conclusion, since it might be unwarranted. Another option (my preferred one, but more time-consuming) is to do something along the lines of data-driven modeling: take some properties of the real data (topology of the city network, distribution of ride requests, etc.), simulate the model on top, and try to at least qualitatively match the empirical distribution of shared trips, to get a hint on the parameter values (epsilon, etc.) that would correspond to NYC and Chicago, and then actually determine whether the city is close to a transition line and would thus benefit from a ‘moderate increase of financial incentives’. Without a closer link between data analysis and the model dynamics, I feel that the conclusions drawn from the perceived two regimes in Fig. 5 cannot really extend to specific suggestions on how to increase shareability in real-world cities.

Authors’ response:

We thank the reviewer for bringing this point to our attention.

We agree with the reviewer that our stylized model produces clearly distinct regimes of ride-sharing adoption, which may be much more difficult to observe in the empirical data. Various sources of heterogeneity (e.g. heterogeneous user preferences, heterogeneous origin-destination demand distributions, heterogeneous city topologies) may modulate ride-sharing decisions and blur the transition between the two regimes, yet leaving the underlying mechanisms outlined in our model unaltered.

To demonstrate (i) the robustness of our results and (ii) to show how heterogeneity qualitatively impacts the presence of disparate regimes of ride-sharing adoption we have prepared new and more detailed simulations of our model under heterogeneous user preferences, as typically found in real cities (see Fig. 4d in the revised manuscript). The simulation results demonstrate how this type of heterogeneity may explain a spread of adoption across the different sharing regimes (forming both a horizontal and a linear branch in the sharing adoption) and at the same time reproduces the observations from New York City qualitatively. The larger fraction of zones with high ride-sharing adoption in New York City thus suggests that the current financial incentives are largely sufficient to overcome the *average* inconvenience preferences except for a few zones in the low sharing regime. We take this to imply that other zones are likely also close to the transition line, assuming that the preference distribution is similar there. This interpretation seems consistent with the observed flat

correlation of income and sharing behavior across the zones in New York City (see replies to reviewer 1 above, Fig. 2). This reasoning may not be correct for the airports in New York City due to the special circumstances compared to normal intra-city traffic. In Chicago, we observe more zones close to the low sharing regime. Here, it is possible (and consistent with our qualitative model predictions) that the incentive values are further from the transition line.

With our statement cited by the reviewer, we intended to convey the idea that the change in ride-sharing adoption is likely not proportional to a change in financial incentives. In this line of thought, we absolutely agree with the reviewer that a quantification of required threshold financial incentives to stimulate the transition towards high ride-sharing adoption would be of high value in real-world applications.

To determine this threshold, precise knowledge of users' actual preference parameters and utility functions were required, or alternatively, would have to be estimated from a parameter fit through various simulation runs for different constellations. Unfortunately, the computational complexity of the needed real-city scale simulations would require long-term simulation runtimes even on super-computers for three reasons: (i) intrinsic algorithmic complexity of the matching problem, (ii) increasing equilibration times of the replicator dynamics for real city configurations, (iii) repeated simulation runs for parameter scans and robustness analysis. *Please note that reviewer 1 inquired about the feasibility of extended simulation scale too, hence we briefly repeat parts of our reply here:*

- (i) *Intrinsic algorithmic runtime hurdles:* Real-city scale simulations may have hundreds of different destinations and hundreds of simultaneous shared ride requests as observed at some locations in the empirical New York City or Chicago trip records. Together with the intrinsic cubic scaling of the perfect matching algorithm in the number of shared ride requests, simulations at this scale quickly become intractable even on super-computers.
- (ii) *Increasing equilibration times:* On the one hand, the number of required Monte Carlo samples to compute players' utility in our model increases strongly with the number of players and different destinations. We have to accurately sample all configurations of different ride-sharing decisions and destinations of the S players according to the given origin-destination distribution for all player types. On the other hand, heterogeneity

of the average distances and origin-destination distribution in real cities translates into heterogeneous equilibration timescales per player type, thus multiplying the total equilibration as all players' decisions may have to (re)equilibrate multiple times when slower players change their decisions. Overall, we expect real-city scale simulations to significantly increase the timescales and game repetitions required to equilibrate the replicator dynamics.

- (iii) *Multiple simulation runs*: To fit our real-city scale simulation to empirical trip records we would need to systematically repeat the simulation runs for various parameters (e.g. inconvenience, detour preferences) to then robustly estimate the critical financial incentive at which the transition towards high ride-sharing adoption occurs. Such variations would further multiply the increases in runtime sketched above.

Owing to these computational limitations, a data-driven estimate of the critical financial incentive is unfortunately infeasible.

As we cannot conclusively identify how close each city or zone is to the transition line predicted by our model (also because the transition is likely blurred out in the empirical data due to heterogeneity of user preferences), we have more carefully explained this idea, the limitations and options to identify the phenomenon from empirical data and clarified the discussion of the implications in the revised manuscript.

Reviewer comment:

Is there an empirically motivated reason to choose the value $u_{\text{single}} = 10$ for the single-ride utility? (beyond ensuring positivity of Eq. 5 as stated in Methods). Are there qualitative differences in the behavior of the replicator dynamics as a function of u_{single} ? (There's some mention of this around Eq. S5 in the SI where the authors mention that the infinite limit corresponds to a continuous replicator dynamics.) Perhaps it would be useful to mention somewhere that the regimes of adoption are robust to the choice of u_{single} , and also mention how u_{single} could be related to some measurable quantity (i.e. ride price).

Authors' response:

We thank the reviewer for their valuable question about the choice and impact of utility parameters. There is no empirical motivation for the choice $u_{\text{single}} = 10$, however, our

FIG. 5. Robustness of model results across a broad range of utility values of u_{single} in all regimes of ride-sharing adoption.

results are robust for variations of this parameter.

Since the decisions are driven by the utility difference between single and shared rides, the utility of a single ride does not impact the equilibrium outcomes (zero utility difference). The parameter u_{single} only determines the timescale of equilibration in the replicator dynamics. The change of user behavior per time step approaches 0 in the limit of $u_{\text{single}} \rightarrow \infty$, effectively resulting in an Euler-step of the continuous replicator equation $\frac{d\pi}{dt} = \pi(1-\pi)\Delta u$ with time step $\Delta t = 1/u_{\text{single}}$. Too small of a value may make the system more susceptible to fluctuations from inaccurate estimation of the average utilities due to an effectively too large step size (see also our reply to reviewer 1, Fig. 4).

In Fig. 5 we demonstrate the robustness of the model predictions for a broad range of utility values u_{single} . Independent of the phase of ride-sharing adoption (compare different values of ζ/ϵ) the choice of u_{single} does not impact the equilibrium outcomes.

In real settings, the value of u_{single} quantifies the utility of a single ride, taking into account the benefit of being transported, but also the ride price, opportunity cost for the trip duration and overall convenience of the transportation service.

We have incorporated Fig. 5 in the SI and added a brief discussion about the impact of choice of utility parameters to the method section of the revised manuscript.

Reviewer comment:

Figure 3: The labels for strategies at the bottom of Fig. 3 b-e are somewhat unclear (inner share, outer share, etc.). I guess the associated incentive plots correspond to the incentive values averaged over subsets of modeled users in certain areas (inner, outer) with given

chosen behaviour (share, single, etc.), right? Further, the ‘mixed’ strategy (in subplots d,e) corresponds to utility $u_{\text{share}} = 0$, right? Perhaps all of this should be written more clearly, at least in the caption. Also, in subplot b, why are the strategies ‘inner share’ and ‘outer share’ repeated two times?

Authors’ response:

We thank the reviewer for pointing out this potential source of confusion.

The reviewer is absolutely correct in their interpretation. The labels "inner" and "outer" in Fig. 3b-e correspond to the inner and outer rings of the stylized city topology. The labels "share", "single", or "mixed" describe those nodes on the inner or outer ring that adopt either a dominant single or sharing strategy, or a mixed strategy where they alternate between single or shared rides with a given probability (subplots d,e). Each subplot shows the incentive balance for a single node with the given characteristics, though the incentive balance is identical across nodes in the same relative position to the origin with the same sharing adoption. In case of a mixed equilibrium strategy, individuals are indifferent between the expected utility derived from a single or a shared ride. In this situation $\Delta u := u_{\text{share}} - u_{\text{single}} = 0$, as correctly pointed out by the reviewer.

In the revised manuscript, we have extended the explanation of these elements of the figure and explicitly labeled the destination nodes for which we illustrate the decomposition of utility in subpanels b-e, which allows for an unambiguous mapping.

Reviewer comment:

Figure 5: Homogeneous vs. heterogeneous patterns of ride-sharing in the high-sharing / low-sharing regimes. Subplots c-f seem to show this dichotomy in the distribution of share-ride requests depending on what regime the location is on. (A very interesting find! Particularly when comparing locations with the same request rate s .) Still, just looking at the color pattern of subplots c-f doesn’t convince me that e.g. LGA is more heterogeneous than East Village. Could you add some other quantitative measure of this spatial heterogeneity / lack thereof? Plus some measure that helps convince us of the statistical significance of the difference in heterogeneity between locations in both regimes? This find is clear in the idealized cities of Figs. 3-4 and the SI, but much less so in the data for NYC and Chicago.

Another comment on the comparison of spatial homogeneity and heterogeneity between sharing regimes: While comparing East Village and LGA in the NYC dataset makes sense since they have similar values of s , comparing Austin and O'Hare in the Chicago dataset is a bit misleading: could it happen that Austin is more homogeneous simply because there are less rides per unit of time (i.e. lower s)? (and again, even then, the pictures almost look similarly heterogeneous to me) A better measure of spatial heterogeneity might help clarify this issue.

Authors' response:

We thank the reviewer for their question about spatial patterns in the different regimes of ride-sharing adoption.

We agree with the reviewer that the patterns observed in New York City have a similar variance. The patterns of ride-sharing adoption predicted by our model in the low sharing regime are strongly dependent on the network topology (see Supplementary Figure 10, 11 and 13) and the spatial distribution of preferences driving the symmetry breaking, i.e. which nodes are more likely to stop sharing first. Quantifying the variance of ride-sharing adoption for different zones, we indeed find very similar values for zones in the high and low sharing regimes. This suggests that the observed spatial patterns are dominantly driven by existing spatial heterogeneities. Relatively high fractions of shared rides from and to zones like Crown Heights North with a relatively low income support this conclusion, though there is only a weak overall correlation between income and sharing adoption in New York City (see Fig. 2 and our reply to reviewer 1).

For Chicago, we compare the two zones with different request rate since there are no zones in the high sharing regime at higher request rate except the large downtown zones with much larger demand. The reviewer is correct that the patterns will not be as directly comparable as for New York City, however the qualitative difference in ride-sharing adoption, i.e. in the relative fraction, not the total number, of shared ride-requests, is visible.

We intended the stylistic city networks in Fig. 5 in the manuscript as a reference to the different model regimes, not necessarily to the spatial patterns. In the revised manuscript, we have modified and amended the presentation in the main text and the figure to clarify this intention and added a brief discussion of the above points.

Reviewer comment:

In the references to the SI (text ‘see Supplementary Information for details’ throughout the manuscript), it would be good to have an explicit reference to the SI section the authors refer to, to make it easier for the reader to find the corresponding information.

Authors’ response:

We thank the reviewer for their comment and have included specific cross-references to Supplementary Notes, Figures and Methods in the revised version of the manuscript.

Reviewer comment:

There’s some mismatch of references to the main text in the SI and the main text itself. For example, in the caption to Fig. S4 there’s a reference to Fig. 4 d in the main text (the subplots only go up to c). Another example are the references to Fig. 5 subplots in the main text (page 6), which seem to be mixed (i.e. referring to heterogeneities in Fig. 5 c, instead of d?). There might be some others I didn’t catch, so I suggest a careful check of figure references, etc.

Authors’ response:

We thank the reviewer for spotting this mismatch. We have carefully checked and updated cross-references in the revised version of the manuscript.

REPLIES TO THE COMMENTS OF REVIEWER 3

Reviewer 3 considers the manuscript and model “interesting” and “novel”. They remark on the assumptions made in deriving our minimal model and request a more detailed discussion on these and further technical aspects regarding the simulation and data analysis.

Reviewer comment:

The authors present a study of the dynamics of ride-sharing adoption in the context of on-demand transportation systems. The authors first model the problem at a theoretical level as an anti-coordination game, then apply the model to a simple synthetic urban network to uncover discontinuous transitions in the system dynamic. The most interesting part of the paper is the last, in which the authors show that those transitions can be observed – at least to some extent – also on large-scale ride-sharing data sets collected from NY and Chicago.

Authors’ response:

We thank the reviewer for their precise summary of the manuscript and their overall interest in our work.

Reviewer comment:

While I find the paper interesting and the model novel, I think the authors can further improve it under several respects.

Authors’ response:

We thank the reviewer for emphasizing interest in and novelty of our research and inviting us to further improve it.

Reviewer comment:

First, the model is based on strong assumptions, that should at least be discussed. In particular, the model is based on the non-stated assumption that, when taking a decision on whether requesting a single or shared ride, the user has "perfect knowledge" of the expected probability of actually sharing the ride, not only a general sense, but a very detailed estimate based on the selected destination. I doubt that this is the case in practical scenarios. Notice

that the probability of sharing the ride depends also on behavioral choices of other users, so in a sense the model assumes that users have a good sense of what all other users in the ride sharing market typically do. This is very rich knowledge that I believe an operator like Uber or Lyft might have based on millions of observations. However, I do not find realistic the assumption that a regular user has access to this knowledge as well. Would the model work also in the context of bounded user knowledge?

Authors' response:

We thank the reviewer for raising this important point.

To avoid any misunderstanding, we explain the idea of our approach, modeling learning of the customers via the replicator equation. In particular, we do not play one-shot games. As such, a single customer does not have any knowledge of the decisions or utilities of other customers. A single customer only has exact knowledge of their own utility *resulting from these decisions*, which they learn over time. In a simple example setting (not specifically analyzed in our work) this would correspond to the fact that a user may not know the reason for other people to take the bus, but learns over time how full the bus usually is and, based on this information, may decide to walk or bike instead.

The expected utility a group of similar customers (identified by their destination) derives from sharing in a single time step of the simulation is computed as the average utility over multiple games (e.g. these customers use the service for a week and then adapt their behavior). Over time, all (groups of) customers converge to an equilibrium state of sharing adoption. These equilibrium states correspond to local equilibria of the one-shot game. As initial state we start the simulation close to the no-sharing state. Varying initial conditions does not result in different states of ride-sharing adoption in our simplified model topology. However, the existence of symmetry breaking and a discontinuous transition between adoption regimes suggests the possibility of more complex dynamics.

In some situations customers may not be able to evaluate their own utility accurately (due to uncertainty, random fluctuations etc.), potentially leading to imperfect decision-making. To demonstrate the robustness of our results also under these conditions we add noise to the realized utility of shared rides. *Please note that reviewer 1 inquired about the impact of imperfect decision-making on our results, hence we repeat parts of our reply here:* Taking $X \sim \mathcal{N}(0, \sigma^2)$ to be a normal random variable with mean zero and variance σ^2 , we realize

the deterministic utility increment Δu as described in the manuscript and multiply it by a factor $(1 + x_{i,n,t})$ where $x_{i,n,t}$ denotes the realization of random variable X for rider i in game n at time t . The standard deviation σ acts as a control parameter for how different an individual might perceive the utility of a shared ride, or how strong external stochastic influences are. The ride-sharing adoption evolves based on the estimated expected utility from 100 realizations (e.g. the experience of a group of similar users over a week). The users thus naturally filter out large variances but do not arrive at a perfect estimate of their expected utility.

For different noise amplitudes we consistently reproduce the average adoption behavior without noise (compare Fig. 4 and also our response to reviewer 1), demonstrating the robustness of our findings also for (i) strongly heterogeneous perception in the benefit of shared rides in the population and (ii) for stochastic service quality such as travel time.

We have added these additional simulation results to the Supplementary Information and clarified the description of the model, also explaining the assumptions of user knowledge, in the revised manuscript and the Methods section.

Reviewer comment:

Authors assume that matching is done based on the maximum number of saved miles. However, I hardly believe that this would be the parameter used by a mobility operator, which would probably prefer to maximize the total number of riders served. In fact, in a highly competitive market like NYC/Chicago, operators might want to increase their market share, i.e., total number of served customers. In this context, that would translate in optimizing for maximum number of matched rides, rather than maximum number of saved miles.

Authors' response:

We thank the reviewer for emphasizing the importance of appropriate incentives among the different market participants to correctly model the overall market dynamics.

We agree with the reviewer that different types of market participants – mobility service provider, competitors, customers, regulators – may all have different incentives. Opposing long-term objectives regarding the market equilibrium (e.g. desired market size, monopoly position in the market) may cause complex dynamics during the transient (e.g. competition among mobility operators, rapid expansion strategies), being reflected, among others, in the

service provider's matching strategies (e.g. minimum distance, maximum profit, maximum growth rate). In our work, we attempt to isolate the customer perspective from this intricate complex of market participants and their incentives.

We focus on a minimal model of ride-sharing adoption in equilibrium conditions, rather than a description of the transient dynamics. To isolate the customers' decision for or against adopting ride-sharing in this model, we do not consider the larger market interactions or potential feedback loops with the provider. Importantly, all S customers are always served, independent of requesting single or shared rides. We chose the matching scheme to minimize distance driven since (i) the commonly provided motivation for ride-sharing is increased sustainability of urban transportation (i.e. reduced emissions due to reduced total distance driven), (ii) for an equilibrated market minimizing distance driven is consistent with a profit-maximizing service provider and (iii) it maps to an efficiently solvable mathematical problem (perfect matching on a graph).

A different matching scheme would modify the expected detour and inconvenience for customers requesting a shared ride (e.g. by bounding the allowed detour from above to avoid unhappy customers). However, the mechanisms captured by our model would remain unchanged. Thus, the same qualitative results are expected with only a quantitative change of, for example, the critical incentive parameters for the transition to full sharing or the sharing adoption patterns in the low sharing regime.

We have clarified this assumption in the revised manuscript.

Reviewer comment:

In the data-based analysis, the authors filter out what they call "infrequent" destinations, using a threshold value of 100 trips. The authors mention that this is done to avoid fluctuations. While this is reasonable, I would like to see a robustness analysis on how the value of the threshold used for filtering might impact the results. One of the reasons for doing this is that, while most of the trends highlighted by the authors in the model and simulations can be observed on real data, the "bifurcation" trend – which is actually one of the main findings of the model analysis – can only be hinted to by looking at the data. Looking at Figure 5, the trend is only barely visible in NY, with very few dots concentrating along the horizontal orange line. Even less in Chicago, especially looking at Supplementary Figure 5, where the few dots (no more than 5) close to the orange line seems to be outliers of a general sublinear

trend (see inset in the same Figure), rather than delineating a clearly separate trend. At the very least, the authors should show that these "hints" to bifurcation observed in the big data analysis are preserved and not an artifact of choosing a specific value of trips for filtering out infrequent destinations.

Authors' response:

We thank the reviewer for their valuable remark on the threshold value of 100 trips and the robustness of the bifurcation trend. The threshold was chosen to avoid fluctuations in the fraction of shared rides from zone pairs which have been requested less than 100 times throughout all of 2019. Normalized to an effective per minute request rate all zones for which the threshold value applies, accumulate around a request rate of approximately 0 rides/min.

Typically, the origin-destination zone pairs subject to this threshold correspond to destinations with low population density according to the US Census Bureau. For example, in Fig. 5c rides from 'East Village' to 'Governor's Island/Ellis Island/Liberty Island', 'Crotona Park' (a public park in South Bronx), 'Jamaica Bay' (a bay of marshy islands), 'Marine Park/Floyd Bennett Field' (a former airfield), 'Rikers Island' (a jail complex) or 'Saint Michaels Cemetery/Woodside' (a cemetery in Queens) are requested fewer than 100 times per year and thus grayed out.

We note that these rides are still included in the overall sharing fractions shown in Fig. 5a, and only excluded when computing detailed origin-destination resolved sharing fractions (as displayed in Figs. 1, or 5c,e). Fig. 5a,b hint at bifurcation points of approx. 5.5 rides/min in New York City and 2.75 rides/min in Chicago, corresponding to roughly 2 million and 1 million annual ride requests, respectively. Hence, our results would be qualitatively robust even for thresholds up to several thousands of rides per year.

Moreover, the bifurcation trend produced by our model is robust in more realistic settings, e.g. reflecting real cities with diverse ride-sharing user base with heterogeneous user preferences. New and extended simulations of our model considering multiple player types with different convenience preferences confirm the persistence of the discontinuous phase transition per player type in the high demand limit (compare also our reply to reviewer 1 regarding this question). Superimposing multiple origins with heterogeneous distributions of user preferences provide an aggregate view of the ride-sharing adoption in a real city (compare Fig. 4c,d in the revised manuscript), and robustly produce (i) the bifurcation trend

between regimes of low- and high-sharing, and (ii) more accurately reproduce the empirical ride-sharing adoption observed from the trip records in New York City.

We have clarified the use and impact of the threshold in more detail in the Methods section in the revised manuscript. Furthermore, we have included the new simulation results on heterogeneous user preferences in the revised version of the main manuscript and provided additional simulations in Supplementary Note 4.

Reviewer comment:

It would be also useful to add a plot showing the evolution of $\pi(d, t)$ with epochs, to assess that the system actually reaches an equilibrium, and the speed of convergence to equilibrium.

Authors' response:

We thank the reviewer for raising this important methodological point which we used to refine and quantify the accuracy of our model results.

We agree with the reviewer that some parameter combinations, e.g. close to the transition line, are subject to stronger fluctuations and require longer simulations to equilibrate. In the newly added Supplementary Figure 18 of the revised Supplementary Information we show the time evolution of the replicator dynamics (see below for details).

In our model analyses we focus on the equilibrium properties of the ride-sharing (anti-) coordination game. To obtain these values we follow the following procedure: we discard the initial transient of the dynamics (4000 time steps), quantify the degree of temporal fluctuations along the trajectory $\pi(d, t)$ in the following 1000 time steps. If the degree of fluctuations exceeds a threshold value (e.g. due to a continued trend), we continue to evolve the trajectory by another 5000 replicator steps and repeat the procedure until we reach a stationary state. We then estimate the stationary points $\pi^*(d)$ by averaging over the last 1000 time steps. Following the reviewer's comment, we have repeated our simulations with stricter thresholds, smaller value of $u_{\text{single}} = 4$ and a minimal evolution time of 20000 replicator time steps of 100 game repetitions each to achieve a more accurate representation of the equilibrium state, especially close to the transition line. We note that the new results do not change our qualitative conclusions or model predictions.

In Fig. 6 shown below we illustrate our approach and quantify the degree of fluctuations (due to imperfect estimation of the expected utilities) around the estimated stationary values

FIG. 6. Equilibration of replicator dynamics in the ride-sharing (anti-)coordination game. **a**, Phase diagram of equilibrium sharing fraction. **b**, Temporal fluctuations in sharing fraction. **c**, Time evolution of sharing fraction for parameter constellation with longest equilibration timescale. Inset shows part of trajectory used to determine value of stationary point (gray, dashed). Shading illustrates 1 percentage point fluctuation corridor. **d**, Transient dynamics of the sharing adoption for all destinations in the stylized city topology for the parameter configuration in panel **c**.

of $\langle \pi^* \rangle$ for the phase diagram shown in Fig. 4 in the main manuscript. Fig. 6b shows the standard deviation $\sigma_{\langle \pi^* \rangle}$ of the sharing fraction over time along the part of the trajectory used to estimate the equilibrium sharing fraction. For none of the $(S, \epsilon, \zeta, \xi)$ -tuples the sharing fraction fluctuates by more than 1 percentage point around the estimated equilibrium value, hinting at a stationary equilibrium state. We observe that the vast majority of trajectories is much better equilibrated with negligible deviations from the estimated value $\langle \pi^* \rangle$ over time. Only in the low-sharing regime and in the vicinity of the transition towards the high-sharing regime stronger fluctuations emerge, as expected.

In Fig. 6c,d we show an exemplary trajectory for the $(S, \epsilon, \zeta, \xi)$ -parameter tuple with longest equilibration timescale to reach the required equilibration threshold (a parameter configuration at the transition to the high sharing regime). All other points reach their equilibrium values (up to the allowed degree of fluctuations) on shorter timescales.

As discussed with reviewer 2 the characteristic timescale of equilibration in the replicator

dynamics depends not only on the choice of preference parameters, but also on the order of magnitude of the riders' utilities determined by the value of u_{single} . The order of magnitude in utility effectively controls the adaptation of $\pi(d, t)$ between successive time steps t and $t + 1$, and may be interpreted as the Euler step in the continuous-time formulation of the replicator dynamics. As shown in Fig. 5 of this reply the qualitative results of the model are robust across a broad range of utility parameters. Hence, the timescale of equilibration may be fine-tuned by appropriate choice of u_{single} without altering the equilibrium outcomes.

We have added Fig. 6 to the Supplementary Information to illustrate the time evolution and the equilibration of the replicator dynamics and added additional information to the Methods section in the revised manuscript.

Reviewer comment:

The authors should clarify whether the "shared ride" flag in the NY and Chicago data set refers to the fact that the ride sharing option has been selected, or otherwise to rides that have actually been shared. My understanding is that the first option holds, at least for Chicago. If so, is there any information in the data sets on which of the requests have actually been performed in ride sharing mode? If available, this information would be very useful to attempt to further validate the model, i.e., comparing the ratio of actually shared rides vs total shared requested rides.

Authors' response:

We thank the reviewer for pointing out the ambiguity in the data description of the publicly available trip records for New York City and Chicago. The reviewer is correct that the "shared ride" flag in the data sets does not imply actually matched rides.

For New York City, the data set does not contain separate information about shared ride requests and actually matched rides. For Chicago the shared ride flag encodes whether a customer agreed to a shared trip with other people, independent of whether the trip has actually been matched or not. Additional data provides information on the number of matched rides, defined as the length of the trip series from the time the vehicle was empty until the time it was empty again, counting shared rides as successfully matched even if vehicle passengers have never actually spent parts of their rides in the same vehicle at the

same time. This definition tries to quantify the amount of "deadheading" – driving without passengers – and takes the perspective of the driver rather than that of the rider, as we do in our analysis. In our model riders are subject to detour and inconvenience effects only if they are simultaneously matched into the same vehicle. Hence, the available data is unfortunately not sufficient to quantitatively estimate the ratio of actually shared rides vs. total shared ride requests as defined in our model.

We have added a discussion of these aspects to the revised version of the manuscript.

Reviewer comment:

In the SI, page 11, there is a reference to Figure 11, which should be Figure 10 instead.

Authors' response:

We thank the reviewer for pointing out this mismatch. We have carefully checked and updated the references in the revised version of the manuscript accordingly.

Reviewer #1 (Remarks to the Author):

I thank the referee for their thorough consideration of my thoughts and suggestions.

While the model is thoroughly explored and has some realism, I am ultimately not convinced by the data. Even with heterogeneity in the data, shown in Figure 4d of the main text, we should see a more significant division between the high and low sharing regimes. While the main text says "Sharing decisions for New York City and Chicago (blue dots) accumulate on two branches corresponding to the predicted high- and low-sharing regime as a function of request rate" the SI instead shows no significant accumulation on two branches - at least to me. I feel that statements like that should be toned down. Your additional evidence presented in the SI is certainly helpful, it is not entirely conclusive either.

The reason for the differences between model and data are in part the usual suspects: topology simplification, and the heterogeneity of human decisions. Namely, the road topology differs significantly from the model (although I understand the model cannot be applied to a realistic topology due to data computer limitations) as well as how people make decisions, which adds noise to the system in a way not thoroughly captured by the model (even in Fig. 4, for instance, I see an accumulation of data on 2 branches, which we do not see in data). Furthermore, data limitations like finite request rates make it even more of a challenge to compare the model and data. All this is to say, I am not convinced the model does not describe how people tend to behave, but I am not convinced that it does either. Instead an even simpler model might be able to explain the broad conclusions here.

Reviewer #2 (Remarks to the Author):

I thank the authors for a careful and thorough revision of their manuscript. They have made additions and changes to the paper that address all of my previous concerns. In particular, the analysis of the model in the case of heterogeneous preferences (revised Fig. 4), as well as a more detailed description of ride-sharing dynamics in the NYC and Chicago empirical data (text related to revised Fig. 5), decrease my concern that the model might be too idealized to draw conclusions relevant to real-world data. I find of great interest the fact that heterogeneous preferences lead to a mixed state where different segments of the population may correspond to either the high- or low-sharing states. The additional details on the robustness of the simulations to parameter changes are welcome as well (Fig. 5 in rebuttal letter, and related text in SI). Finally, I appreciate the new Discussion section, which is far more precise in stating the scope of the model and its limitations in terms of applicability, highlighting the potential for future research in this direction. I believe the manuscript is now suitable for publication in Nat. Comms.

Reviewer #3 (Remarks to the Author):

The authors have done a good job overall at addressing not only my comments, but also those of the other reviewers. I still have one pending concerns from my side, namely my comment about the choice of the metric used to perform max matching (saved miles).

Sorry to insist on this, but I think that the main contribution of the paper is the identification of this bifurcation dynamic related to incentives for ride sharing. While more accurately quantify the right amount of incentives to be offer to reach the "high sharing" area might be out of scope for a number of reasons, as the authors comment in their replies, what I think is key to assess is: does this dynamic exists also in real-world market conditions? A positive answer to this question would attest to the significance of this study.

In real world, company like Uber and Lyft *do not use km saved to build shared rides*. I can assert this with a high deal of confidence. They simply do not care about saved miles (this is just something they use for PR reasons), they just want to have more passengers on their cars. This aggressive logic to serve customers is probably not even rational from a business viewpoint, as the authors correctly observe, yet, it is what the operators do.

In view of the above, I would recommend the authors to run another run of simulations using non-

weighted matching. If the authors want to avoid complications related to bounds on detour distance, etc, they might simply assume that a link between two trips exists only if saved distance is >0 . (even though I am quite sure TNCs would serve a shared ride also with save distance <0). I am quite confident that the bifurcation behavior would persist, and the results would be even more interesting because would be based on a matching mechanism closer to the one actually used by TNCs.

Replies to the comments of the reviewers

on the manuscript

Incentive-driven transition to high ride-sharing adoption

by David-Maximilian Storch, Marc Timme and Malte Schröder

GENERAL REPLIES

We thank all three reviewers for their detailed comments and positive evaluation. Overall, the reviewers consider our first revision “careful and thorough” and generally agree that it addresses all previous comments. Reviewer 2 recommends publication in Nature Communications as is. Reviewer 1 and 3 have two remaining suggestions that we have addressed in our second revision.

Reviewer 1 asks us to more carefully phrase the results of our analysis in the context of bridging between observations from our model and the empirical data. We have carefully revised the manuscript and now more precisely emphasize which conclusions arise from the model, which from the data alone, and exactly which aspects of the data are predicted by the model compared to those that might be coincidental.

Reviewer 3 requests one more set of simulations to confirm the robustness of our results for different matching strategies by the provider. We have tested two different strategies suggested by the reviewer and find qualitatively the same regimes with low, partial, and high ride-sharing adoption. However, the quantitative transition point varies depending on the matching strategy.

We have revised the manuscript and the Supplementary Information based on these suggestions and provide a detailed response to all referee comments below.

REPLIES TO THE COMMENTS OF REVIEWER 1

Reviewer 1 suggests we further clarify the discussion and more precisely state the main results, the scope of the model, and its limitations in terms of applicability.

Reviewer comment:

I thank the referee for their thorough consideration of my thoughts and suggestions.

While the model is thoroughly explored and has some realism, I am ultimately not convinced by the data. Even with heterogeneity in the data, shown in Figure 4d of the main text, we should see a more significant division between the high and low sharing regimes. While the main text says "Sharing decisions for New York City and Chicago (blue dots) accumulate on two branches corresponding to the predicted high- and low-sharing regime as a function of request rate" the SI instead shows no significant accumulation on two branches – at least to me. I feel that statements like that should be toned down. Your additional evidence presented in the SI is certainly helpful, it is not entirely conclusive either.

The reason for the differences between model and data are in part the usual suspects: topology simplification, and the heterogeneity of human decisions. Namely, the road topology differs significantly from the model (although I understand the model cannot be applied to a realistic topology due to data computer limitations) as well as how people make decisions, which adds noise to the system in a way not thoroughly captured by the model (even in Fig. 4, for instance, I see an accumulation of data on 2 branches, which we do not see in data). Furthermore, data limitations like finite request rates make it even more of a challenge to compare the model and data. All this is to say, I am not convinced the model does not describe how people tend to behave, but I am not convinced that it does either. Instead an even simpler model might be able to explain the broad conclusions here.

Authors' response: We thank the reviewer for their detailed and well-considered reply.

Our model shows that there are qualitatively different regimes of ride-sharing adoption (low, partial, high). In the limit of high demand, the partial sharing regime disappears, and the system abruptly transitions between the low and high sharing regime when financial incentives change. Heterogeneous customer preferences in the same model result in more diverse adoption behavior with each subset of customers (identified by their preferences),

exhibiting the transition between high and low sharing explained above. Overall, we find a spread of sharing adoption behavior in heterogeneous settings bounded by the qualitative behavior in the full sharing (a linear increase of the number of shared requests with the total number of requests) and the low sharing regime (constant number of shared requests as the total number of requests increases) observed in the simple model setting. As suggested by all reviewers in their previous comments and now included in the Supplementary Information, additional simulations demonstrate that the above results are robust even under a variety of more complex settings or changes in the details of the model, such as different matching algorithms, different request distributions, or under uncertainty and noise influencing the users' decisions.

In this sense, we agree with the reviewer that "an even simpler model might be able to explain the broad conclusions". Indeed, this constitutes our main finding: *the qualitative ride-sharing adoption behavior results from the interplay between financial and inconvenience (incl. detour) incentives alone*. To conduct our simulations and analyses, we have in the main manuscript aimed to make the simplest possible choices for the other parameters without adding spurious assumptions to keep the model as simple as possible.

In all settings we have analyzed, we also find that, for a given user group defined by fixed preferences, a small increase of the financial incentives may push this group from the low to the high sharing regime, due to the discontinuous transition between the two regimes. In a setting with heterogeneous user preferences, this transition remains per individual user group while the overall adoption is a mix between all user groups and likely increases continuously, though likely non-linearly with the financial incentives.

Thus, if the fundamental interactions between the relevant incentives (based on empirical studies) are correctly captured in our model, we expect empirical ride-sharing adoption to follow similar qualitative trends since the specific details do not influence the qualitative behavior. Indeed, we observe the same qualitative adoption behavior in the empirical data with a linear increase of shared rides for low demand and a broader spread of sharing adoption at higher demand. Of course, as the reviewer mentions, the empirical ride-sharing adoption is subject to several additional influences not explicitly captured in our model, such as various aspects of socio-economic heterogeneity, different traffic and street network conditions, or different regulatory conditions. Therefore, we do not expect to (nor do we claim to) predict the *quantitative* values of ride-sharing adoption in different parts of New

York City or Chicago with our model (as also discussed in our previous replies). Instead, our model correctly and consistently predicts the *qualitative* shape of the empirical data.

In sum, we agree with the reviewer that the data do not unambiguously confirm our model. However, (i) the data are fully consistent with our qualitative model predictions, (ii) our main finding suggest that the transition emerges robustly in a wide range of settings and (iii) to our knowledge, no other studies mechanistically explain the observed data on a level as general as our model.

We have carefully revised the manuscript to more precisely state the main results of our model and their connection to the empirical data, including the explicit example stated by the reviewer. In particular, we highlight which conclusions may be drawn from the model, exactly which aspects of the empirical data are predicted by the model and which additional influences we might expect that are not captured in our analysis. We thank the reviewer for their valuable help with refining the presentation of our results.

REPLIES TO THE COMMENTS OF REVIEWER 2

Reviewer 2 considers the revisions to the manuscript and the Supplementary Information to address all points raised in the previous comments, in particular about "conclusions to real-world data." They praise the additional analyses of heterogeneous preferences and the clarified discussion. As a result, they recommend publication in Nature Communications without further changes.

Reviewer comment:

I thank the authors for a careful and thorough revision of their manuscript. They have made additions and changes to the paper that address all of my previous concerns. In particular, the analysis of the model in the case of heterogeneous preferences (revised Fig. 4), as well as a more detailed description of ride-sharing dynamics in the NYC and Chicago empirical data (text related to revised Fig. 5), decrease my concern that the model might be too idealized to draw conclusions relevant to real-world data. I find of great interest the fact that heterogeneous preferences lead to a mixed state where different segments of the population may correspond to either the high- or low-sharing states. The additional details on the robustness of the simulations to parameter changes are welcome as well (Fig. 5 in rebuttal letter, and related text in SI). Finally, I appreciate the new Discussion section, which is far more precise in stating the scope of the model and its limitations in terms of applicability, highlighting the potential for future research in this direction. I believe the manuscript is now suitable for publication in Nat. Comms.

Authors' response:

We thank the reviewer for their positive evaluation and their valuable feedback during the revision.

REPLIES TO THE COMMENTS OF REVIEWER 3

Reviewer 3 considers the revision to address all previous comments and only requests one set of additional simulations to show the robustness of our results with respect to other/more realistic matching schemes.

Reviewer comment:

*The authors have done a good job overall at addressing not only my comments, but also those of the other reviewers. I still have one pending concerns from my side, namely my comment about the choice of the metric used to perform max matching (saved miles). Sorry to insist on this, but I think that the main contribution of the paper is the identification of this bifurcation dynamic related to incentives for ride sharing. While more accurately quantify the right amount of incentives to be offer to reach the "high sharing" area might be out of scope for a number of reasons, as the authors comment in their replies, what I think is key to assess is: does this dynamic exists also in real-world market conditions? A positive answer to this question would attest to the significance of this study. In real world, company like Uber and Lyft *do not use km saved to build shared rides*. I can assert this with a high deal of confidence. They simply do not care about saved miles (this is just something they use for PR reasons), they just want to have more passengers on their cars. This aggressive logic to serve customers is probably not even rational from a business viewpoint, as the authors correctly observe, yet, it is what the operators do. In view of the above, I would recommend the authors to run another run of simulations using non-weighted matching. If the authors want to avoid complications related to bounds on detour distance, etc, they might simply assume that a link between two trips exists only if saved distance is >0 . (even though I am quite sure TNCs would serve a shared ride also with save distance <0). I am quite confident that the bifurcation behavior would persist, and the results would be even more interesting because would be based on a matching mechanism closer to the one actually used by TNCs.*

Authors' response:

We thank the reviewer for their favorable judgement of our revisions and their suggestion to further strengthen our findings by demonstrating the robustness of the bifurcation in ride-sharing adoption for different matching strategies.

In the second round of revisions, we have created new simulations where the service provider offers ride-hailing or -sharing services under two additional matching strategies for shared rides that both reproduce our qualitative results:

- (i) *First-in, first-out matching*: this unweighted matching strategy ignores the requests' destinations and thus also the distance driven. It pairs subsequent shared ride requests irrespective of distance savings potential. This strategy corresponds to an unweighted matching problem, as suggested by the reviewer, and may correspond to a real-world scenario where the provider applies a first-in, first-out matching strategy without batch-processing. The provider makes sure to offer the customer a ride as quickly as possible, ensuring high vehicle utilization.
- (ii) *Batch-processing*: in this strategy, the provider always matches shared ride requests independent of a distance savings potential. However, other than first-in, first-out matching, it batch-processes all S requests within a specific time-window to pair individuals to optimize provider profits. This matching scheme is similar to the one used in the manuscript but without the distance constraint.

Figure 1 in this reply (Fig. S17 in the revised Supplementary Information) demonstrates the robustness of the low-, partial-, and high-sharing regimes of ride-sharing adoption under these alternative matching strategies. For financial discounts insufficient to compensate inconvenience disutilities the number of shared rides becomes constant as the total demand increases (Fig. 1, green triangles). Only if financial discounts overcompensate expected detour and inconvenience effects,

$$\epsilon d_{\text{single}} > \zeta E[d_{\text{inc}}|\text{share}] + \xi E[d_{\text{det}}|\text{share}], \quad (1)$$

we observe the full-sharing regime where $S_{\text{share}} = S$ (Fig. 1, blue triangles). In contrast to the matching scheme discussed in the main manuscript, the detour preference ξ of customers remains important as long as detours may persist, despite better matching options in the high demand limit. The partial-sharing regime (Fig. 1, orange triangles) separates both regimes. These results qualitatively reproduce Figure 4 from the main manuscript.

While the spatial patterns of sharing adoption change slightly under alternative matching strategies (see Fig. 1), the underlying mechanism leading to the emergence of disparate regimes of ride-sharing adoption robustly persists. Users seek to avoid expected detours

FIG. 1. **Robustness of regimes of ride-sharing adoption for alternative matching algorithms.** Matching shared rides independent of distance savings potential compared to single rides reproduces low- (green), partial- (orange) and high-sharing (blue) regimes of ride-sharing adoption. The orientation of spatial ride-sharing patterns follows from symmetry breaking, reflecting how ride-sharing users try to minimize expected detour and inconvenience disutilities. **a** Unweighted matching based on a first-in, first-out principle (pairing of subsequent rides, independent of destination). **b** Batch-based matching to optimize platform profits (pairing rides across all destinations, but aiming to maximize platform profits).

and inconvenience effects from ride-sharing. To minimize these disutilities, only close-by destination nodes adopt sharing strategies with a spatial extension depending on the financial incentives. A symmetry-breaking process selects the sharing destinations (compare insets in Fig. 1). For matching strategies not aimed at minimizing distance driven only one cardinal sharing direction persists under increasing demand, as alternative settings could potentially lead to long detours. Individuals sharing rides in this cardinal direction manage to completely avoid detours. If financial incentives are sufficiently high to compensate for the remaining inconvenience, the partial-sharing regime persists under increasing demand (Fig. 1, orange triangles). If financial incentives are insufficient, the sharing adoption gradually fades out, producing the saturation of the number of shared rides (Fig. 1, green).

We have added a discussion about the impact of platform strategies and matching algorithms to the revised version of the manuscript. We have also included the robustness analysis sketched above in the Supplementary Information. We thank the reviewer again for this valuable suggestion.

Reviewer #1 (Remarks to the Author):

I thank the authors for their work, and believe it should be published in Nat Comm. With regard to their comments,

(i) the data are fully consistent with our qualitative model predictions

- This is true, and ultimately, stronger tests of this model would likely be out of scope anyway.

(iii) to our knowledge, no other studies mechanistically explain the observed data on a level as general as our model.

- While I have shared my reservations about the model, I also agree that it is an elegant way to explain what is uncovered in these data.

For that reason, I would be fine to see this published. I find this work quite interesting.

Reviewer #3 (Remarks to the Author):

The authors have done a good job addressing my further comment regarding matching algorithms. I have no further comments, and I think the paper is ready for publication.

Replies to the comments of the reviewers

on the manuscript

Incentive-driven transition to high ride-sharing adoption

by David-Maximilian Storch, Marc Timme and Malte Schröder

Reviewer 1 comment:

I thank the authors for their work, and believe it should be published in Nat Comm. With regard to their comments,

(i) the data are fully consistent with our qualitative model predictions

- This is true, and ultimately, stronger tests of this model would likely be out of scope anyway.

(iii) to our knowledge, no other studies mechanistically explain the observed data on a level as general as our model.

- While I have shared my reservations about the model, I also agree that it is an elegant way to explain what is uncovered in these data.

For that reason, I would be fine to see this published. I find this work quite interesting.

Reviewer 3 comment:

The authors have done a good job addressing my further comment regarding matching algorithms. I have no further comments, and I think the paper is ready for publication.

Authors' response:

We thank the reviewers for their positive evaluation and their valuable feedback during the revisions.